# Explaining $CO_2$ fluctuations observed in snowpacks

Laura Graham[1] and David Risk[1]

[1]Department of Earth Sciences, St. Francis Xavier University, Antigonish, Nova Scotia, Canada B2G 2W5

*Correspondence to:* Laura Graham (grahamlau7@gmail.com)

**Abstract.** Winter soil carbon dioxide ($CO_2$) respiration is a significant and understudied component of the global carbon (C) cycle. Winter soil $CO_2$ fluxes can be surprisingly variable, owing to physical factors such as snowpack properties and wind. This study aimed to: quantify the effects of advective transport of $CO_2$ in soil-snow systems on the sub-diurnal to diurnal (hours to days) timescale, use an enhanced diffusion model to replicate the effects of $CO_2$ concentration depletions from persistent

winds, and use a model-measure pairing to effectively explore what is happening in the field. We took continuous measurements of $CO_2$ concentration gradients and meteorological data at a site in the Cape Breton Highlands of Nova Scotia, Canada to determine the relationship between wind speeds and $CO_2$ levels in snowpacks. We adapted a soil $CO_2$ diffusion model for the soil-snow system, and simulated stepwise changes in transport rate over a broad range of plausible synthetic cases. The goal was to mimic the changes we observed in $CO_2$ snowpack concentration to help elucidate the mechanisms (diffusion, advection)

responsible for observed variations. On sub-diurnal to diurnal timescales with varying winds and constant snow levels, a strong negative relationship between wind speed and $CO_2$ concentration within the snowpack was often identified. Modelling clearly demonstrated that diffusion alone was unable to replicate the high frequency $CO_2$ fluctuations, but simulations using above-atmospheric snowpack diffusivities (simulating advective transport within the snowpack) reproduced snow $CO_2$ changes of the observed magnitude and speed. This confirmed that wind-induced ventilation contributed to episodic pulsed emissions from

the snow surface and to suppressed snowpack concentrations. This study improves our understanding of winter $CO_2$ dynamics to aid in continued quantification of the annual global C cycle, and demonstrates a preference for continuous wintertime $CO_2$ flux measurement systems.

## 1   Introduction

The global soil carbon (C) pool stores three times the amount of C as the atmosphere. Organic C reserves in high latitude

soil are disproportionately affected by anthropogenic climate change (IPCC, 2013). Careful assessment of the soil C pool and corresponding fluxes in these often snow-covered, high-latitude regions is critical for understanding the future global C cycle, as increasing global temperatures are likely to stimulate soil $CO_2$ emissions (Raich et al., 2002).

    Cold and wet conditions, like snow cover, pose challenges for measuring wintertime carbon dioxide ($CO_2$) fluxes (Liptzin et al., 2009), leading many studies to focus on ecosystem respiration during the growing season. For instance, seasonal variation

in soil $CO_2$ fluxes is not always discussed in meta-analyses of global soil C studies, whether or not wintertime measurements were included in individual studies (Scharlemann et al., 2014). Despite this skewed focus, soil $CO_2$ is still produced throughout

the winter, even at $-7°C$ (Flanagan and Bunnell, 1980; Coxson and Parkinson, 1987; Brooks et al., 1996). In some cases, the insulating snowpack can prevent soils from freezing completely, stimulating soil $CO_2$ emissions (Grogan and Jonasson, 2006; Larsen et al., 2007; Monson et al., 2006). Further, snow is a porous medium where soil $CO_2$ emissions easily pools, complicating measurement techniques. There has been an observed decrease in Northern Hemisphere snow cover and an earlier onset of spring melt since the 1950s as a result of climate change (Dyer and Mote, 2006; IPCC, 2013). Dyer and Mote (2006) indicated that these changes in snow cover are associated with increasing air temperatures and variations in precipitation. Additionally, increasing air temperature results in increased water vapour in the atmosphere (an increase of approximately 7% in water holding capacity of air per $1°C$ warming), generating more intense precipitation events, including snow storms (Trenberth, 2011). Despite decreases, snow covers 44–53% of Northern Hemisphere land area during winter months (Barry, 1992). With the complex interplay between changes in precipitation, temperature, snow cover, and $CO_2$ emissions in recent and future decades, winter soil $CO_2$ measurements are important for accurate estimates of annual $CO_2$ soil respiration (Fahnestock et al., 1999).

There are several methods of measuring $CO_2$ fluxes through snowpacks including the snowpack gradient technique, chamber methods, and eddy covariance. The snowpack gradient technique is a commonly used technique, and, based on Fick's first law of diffusion, uses $CO_2$ concentration measurements through a vertical profile from the soil to the snowpack surface to calculate flux (McDowell et al., 2000; Seok et al., 2009). This technique minimizes disturbance to the snowpack when compared with the chamber method and does not require homogenous terrain, as for eddy covariance. However, the snowpack gradient technique requires many assumptions and cannot easily account for advective transport of $CO_2$ through snowpacks (McDowell et al., 2000; Seok et al., 2009). Measurement frequencies of wintertime $CO_2$ fluxes in past gradient studies have ranged widely, from only twice per winter, to half-hourly (Liptzin et al., 2009). Measurements of wintertime $CO_2$ fluxes recorded at a higher frequency (half-hourly) have shown that wintertime $CO_2$ fluxes can be surprisingly variable, depending more on transport of $CO_2$ than on microbial variation (Bowling et al., 2009; Seok et al., 2009). This variability presents a problem, because it obfuscates any biological sensitivity to environmental drivers. Under what conditions does the soil microbial community thrive over-winter? This is difficult to determine if observed variations are caused by abiotic factors. For example, Seok et al. (2009) observed patterns of high temporal variability in wintertime subniveal $CO_2$ flux, ranging from 0 $\mu$mol m$^{-2}$s$^{-1}$ to 1.2 $\mu$mol m$^{-2}$s$^{-1}$ during a period of relatively steady soil conditions (temperature, moisture) below $0°C$. Steady soil conditions therefore rule out a microbial driving force when variable fluxes were observed. As for advective transport, it does not increase production of $CO_2$ in soils, but changes the rate of exchange (Bowling and Massman, 2011).

Although we accept the assumption that $CO_2$ production occurs in snow-covered soils, there are methodological limitations for quantifying this $CO_2$ production. Transport of this $CO_2$ out of soils into the overlying media, whether snow or open air, is driven by two main mechanisms: diffusion and advection (also known as bulk flow or mass flow) (Janssens et al., 2001; Roland et al., 2015). The mode of this gas transport through snowpacks affects the timing and magnitude of $CO_2$ release to the atmosphere, and will potentially create significant lags between the times of $CO_2$ production and emission. Under calm conditions, it is generally accepted that trace gases are transported out of soils and through snowpacks into the overlying atmosphere via diffusion. Explained by Fick's first law, the background theory of diffusion assumes that trace gas transport out

of soils or through a snowpack occurs vertically, with the magnitude of fluxes determined by the concentration gradient (Seok et al., 2009). Advective transport from wind, however, can also affect the transport of trace gases such as $CO_2$ through porous media like soil and snow (Kelley et al., 1968; Janssens et al., 2001).

Studies are increasingly showing that this non-diffusive (advective) mass transport (i.e. wind) through snow is significant, and must be taken into consideration (Bowling and Massman, 2011; Rains et al., 2016), while considering the appropriate timescale. Advective transport of trace gases through naturally permeable media occurs due to variations in atmospheric pressure at the surface, and have been studied on both high frequency timescales (seconds to minutes (Massman et al., 1995)) and low frequency timescales (barometric (Bowling and Massman, 2011)). These natural advective flows are ubiquitous, and should also be considered on the mid-range timescale of hours to days (Rains et al., 2016). Bowling and Massman (2011) make it clear that wind pumping in the snowpack enhances outward rates of transport. They measured slower bulk air velocities in snow, which fell within the range of $10^{-3}$ to $10^{-2}$ m s$^{-1}$, implying that the contribution of advection to trace gas transport through snowpacks was smaller than that of diffusion. Modelling results from Massman et al. (1997) indicate that advective transport can either enhance or diminish fluxes by a wide range of 1.5% to 25%, and so further studies with field experiment components are required. A more recent study by Bowling and Massman (2011) found enhanced transport of $CO_2$ beyond diffusive transport by up to 40% in the short term, and 8% to 11% when considering the snow-covered season as a whole. The net combined effect of advective and diffusive transport in snow environments on $CO_2$ and other trace gas transport is considered to be an enhancement to diffusive transport. These studies that investigated advective influence on $CO_2$ transport in snow systems encouraged further study in this area, and so we intended to help fill this gap with our study. To do so, we investigated the mid-range timescale of the re-establishment of consistent $CO_2$ concentration gradients in the snowpack after a wind-induced disturbance using both field and modelling methods.

Our overarching objective was to help overcome the methodological limitations of quantifying wintertime $CO_2$ production. Specifically, we aimed to quantify the effects of advective transport of $CO_2$ in soil-snow systems on the sub-diurnal to diurnal (hours to days) timescale, and to mechanistically describe these behaviours using a 1-dimensional advective-diffusive model adapted for the soil-snow-atmosphere system.

## 2 Methods

### 2.1 Continuous automated field monitoring

The primary motivation for establishing our field stations was to determine the relationship between wind speed, snowpack ventilation, and snowpack $CO_2$ concentration. The site selected is on a plateau in a recovering boreal system at North Mountain, Nova Scotia, Canada in the Cape Breton Highlands National Park. Wintertime snow patterns at North Mountain allow for snowpacks of up to 3 m, with the last of the snow melting in May or June, depending on the timing and amount of snow in a given year. Average annual air temperature at North Mountain is 5.1°C (1999–2013). Average winter air temperature is −6.1°C (January–March, 1999–2013). An insulating snowpack is often established before soils have a chance to freeze completely. Therefore, soils often remain above 0°C throughout the winter, and over-winter $CO_2$ production from these soils is very likely

(Grogan and Jonasson, 2006; Larsen et al., 2007; Monson et al., 2006), as soils produce $CO_2$ down to $-7°C$ (Flanagan and Bunnell, 1980; Coxson and Parkinson, 1987; Brooks et al., 1996). Average annual wind speed is 17.3 km h$^{-1}$, with highest wind speeds in the winter (20.7 km h$^{-1}$, January–March, 1999–2013). High winds and variable meteorological conditions (intense snowsqualls, freeze-thaw cycles) create varying snow depths within close proximity (tens to hundreds of m).

Two measurement stations were installed 60 m apart at North Mountain in late 2013, with data collection from 12 November 2013 to 26 March 2014 and 15 April to 29 April 2015. The sites are referred to as NM1 (North Mountain 1: 46°49'7.41" N, 60°40'20.16" W) and NM2 (North Mountain 2: 46°49'9.15" N, 60°40'18.67" W). The key environmental difference between the two sites was the predictably differing snow depth. At each of the two stations, $CO_2$ concentration through the snow profile was measured at three depths (0, 50, and 125 cm from the soil surface) using Vaisala CARBOCAP® Carbon Dioxide

Probe GMP343 sensors. A Campbell Scientific CR3000 datalogger was used at NM1, and a Campbell Scientific CR1000 datalogger was used at NM2 to control the instrumentation, recording values every 30 minutes. To save power and to minimize potential heating impacts, the GMP343 sensors were turned on for 10 minutes preceding measurement, a measurement was taken averaged over 1 minute, and then the sensors were turned off for the remainder of the 30 minute interval. Optics heaters of the GMP343 sensors were kept off entirely, as there was a very limited risk of condensation formation in the relatively

constant temperature environment of a snowpack. This further reduced potential sensor heat from $< 3.5$ W (optics heaters on) to $< 1$ W (optics heaters off). Together, turning the GMP343 sensors off regularly and keeping the optics heaters off at all times minimized any small potential heating impacts of the sensors. Data was collected from the dataloggers at the end of the winter. One BP Solar 50 W solar panel and one Discover D12550 12 V battery was used to power each of the two stations. Snow depth was measured at both stations using SR50A Sonic Ranging Campbell Scientific sensors. A Young Wind Monitor

(Model 05103) anemometer measured wind speed at NM1. Figure 1 gives the general structure of these stations.

To enhance the field campaign, adjustments were made to the NM2 station for winter 2015 by adding additional $CO_2$ measurements throughout the vertical profile. Specific measurements recorded at NM2 include $CO_2$ concentration at 5 cm depth in the soil, soil surface, and at 25 cm, 50 cm, 75 cm, and 100 cm above the soil surface (in the snowpack). We continued to record ambient air $CO_2$ concentration, wind speed, and snow depth. Measurement recording frequency for all measurements

was adjusted to hourly for 2015. The profiler system for the enhanced concentration profile experiment contained two Eosense eosGP (dual channel nondispersive infrared) sensors to measure $CO_2$ concentrations for select time periods over the 2015 winter. A pump within the station enclosure extracted air samples from the various sampling locations via flexible nylon tubing, carrying the air to the sensor. In-snow and in-soil terminal ends of nylon tubing sampled from 550 mL PVC tubes that had openings covered with high-density polyethylene membranes to exclude liquid water. Data extracted from winter 2015 for

analysis ranged from 15 April to 29 April 2015.

## 2.2 Field data analysis

To examine the degree of concentration decrease after wind ventilation started, we focused on periods in which the likelihood of steady state gas transport was maximized (initial winter 2014 experiment). This is an assumption of the snowpack gradient technique, and we assumed that disturbance to the snowpack, including snowfall, results in deviations from steady state (Mc-

Dowell et al., 2000). We extracted data for time periods during which snow depth had not changed more than several cm in the previous 3 days, meaning that there had been no melt or appreciable new snow. To do this, we took the rolling four-hour mean of the snow depth values and found the difference between each set of consecutive snow depth values. We retained the values for which the difference of the rolling mean was $< 0.001$ m. We conducted regression analyses of $CO_2$ concentration

at the three depths and the corresponding wind speeds during these steady-state periods. The ideal situation (the best set of environmental conditions for which a strong negative correlation could be found) was satisfied when winds increased slowly, then abated several hours later (and vice versa). In order to select data where characteristic response patterns of concentration depletion with increasing wind were present, data were further filtered to satisfy the following conditions: 1) the relationship produced a slope $< 0$, i.e. there was a negative relationship between the two variables, and 2) $R^2 \geq 0.1$. Any relationships

that had a strength of $< 0.1$ were discarded to eliminate weak relationships that may have occurred due to highly turbulent winds, overly short-term winds, overly persistent winds, or other mechanisms that would have resulted in significant complexity. Mean $R^2$ values were calculated, divided by site (NM1 and NM2) and height within snowpack (0 cm, 50 cm, and 125 cm). Our data filtering technique was biased towards selecting periods of steady state and negative correlations between wind speed and $CO_2$ concentration. While the criteria seem demanding, in practice they were less restrictive than one might expect, and

nearly one-fifth of all the measured data passed these filters and were included in the final analysis. With the use of our filtering process, our analysis does not represent an estimate of $CO_2$ flux during the snow-covered period.

   We inspected the enhanced concentration profile experiment data (winter 2015) as a time series to analyze the effect of changing wind speed on $CO_2$ concentration at various levels within the snowpack. To quantify the effect of wind on $CO_2$ snowpack concentration, we identified the time periods when an abrupt increase in wind speed resulted in a rapid decrease in

$CO_2$ concentration. These time periods were then used to determine the rate at which $CO_2$ decreased with an increase in wind speed. This was done in order to directly compare the field data with the modelled $CO_2$ data (see section 2.4, Field-model comparisons).

## 2.3   Model development and sensitivity testing

We developed a model to explore the control of three parameters on the $CO_2$ dynamics of a soil-snow system: soil diffusivity,

snow diffusivity after initialization (advective wind intensity), and snow depth. The goal of this model was to use a diffusive model to mimic advective wind events through a snowpack. A previously existing multilayer 1-D soil diffusion model (Nickerson and Risk, 2009) was adapted for the soil-snow system. The exchange of $CO_2$ between layers was determined by Fick's first law, which assumes that gas transport through a diffusive medium is controlled by the concentration gradient, and occurs vertically. Fick's first law is given as follows:

$$F_{CO_2} = -D_{CO_2} \left( \frac{\partial C_{CO_2}}{\partial z} \right),$$

where $F_{CO_2}$ is $CO_2$ flux ($\mu$mol m$^{-2}$s$^{-1}$), $D_{CO_2}$ is $CO_2$ diffusivity within the snowpack (m$^{-2}$s$^{-1}$), and $\frac{\partial C_{CO_2}}{\partial z}$ is the $CO_2$ concentration gradient of the snowpack ($\mu$mol m$^{-3}$). The diffusivity of $CO_2$ within the snowpack can be calculated empirically using snowpack porosity (based on density), tortuosity, the diffusion coefficient of the specific gas under standard temperature

and pressure, ambient pressure, and snowpack temperature (Seok et al., 2009). We tested a range of diffusivities (soil and snow), along with snow depth, but for simplicity, we did not test ranges for individual parameters that are used to calculate diffusivity (e.g. snowpack porosity, tortuosity).

The purpose of the induced change of an increased snowpack $CO_2$ diffusivity was to mimic observed changes in $CO_2$ flux and snowpack concentration. Specifically, the induced increase in snowpack $CO_2$ diffusivity was used to simulate an advective wind event within a diffusive model. With Atlantic Computational Excellence Network (ACEnet) high performance computers, we used model runs to explore the control of each of the three parameters on the $CO_2$ dynamics of the soil-snow system. The three parameters investigated were soil diffusivity ($m^2s^{-1}$), snow diffusivity at step change ($m^2s^{-1}$), and snow depth (cm). The tested range for each of the parameters is given in Table 1.

We initialized the model using a linear $CO_2$ concentration profile through the layers, determined by soil $CO_2$ diffusivity, layer height, and atmospheric $CO_2$ concentration (set at 380 ppm). Each model simulation began with the system in equilibrium state, which means storage flux was set to 1 $\mu$mol $m^2s^{-1}$. We define storage flux here as the change in $CO_2$ storage in the snowpack, analogous to the exchange of $CO_2$ between the snowpack and the atmosphere. Varying numbers of snow layers were added on top of the 100 cm of modelled soil layers with the following distinctions: 1) we assumed that snow has a higher porosity than the underlying soil, therefore the snow layer diffusivities were always set to a value higher than the soil layers, and 2) we assumed that snow does not produce $CO_2$, therefore we removed $CO_2$ production from the snow layers.

Initial condition snow diffusivity was held constant at $8 \times 10^{-6}$ $m^2s^{-1}$ for all simulations. Since snow diffusivity encompasses porosity, density, and tortuosity, these parameters also remained constant for initial conditions for all simulations: we assumed a homogeneous snowpack, and did not test a range of snow diffusivities for initial conditions. To mimic a range of wind events, after initialization, we tested a range of snow diffusivities. Our ten test values for this snow diffusivity, mimicking advective "wind events", ranged linearly from $8 \times 10^{-6}$ (equal to the snow diffusivity at initial conditions) to $9.08 \times 10^{-5}$ $m^2s^{-1}$ (approximately the diffusivity of $CO_2$ in air) (Table 1). We tested a plausible range of soil diffusivity and snow depth values (parameters used for initializing), though these remained unchanged through the "wind event" in each simulation. We tested a range of soil diffusivities to mimic a range of $CO_2$ emission rates out of the soil into the overlying snowpack. A range of snow depths was tested to mimic the natural environment that we tested in the field.

Figure 2 shows an example of the apparent storage flux and corresponding change in snowpack $CO_2$ concentration at every 10 cm, with an induced change in $CO_2$ snowpack diffusivity, which was the mechanism used to mimic an advective "wind event". In summary, to simulate how the modelled diffusive system responds to an advective wind event, the model simulated induced changes in transport rate (snow diffusivity) within the snowpack over a range of plausible synthetic base cases (soil diffusivity and snow depth). We ran the model with all possible permutations of the three parameters given in Table 1.

It is very likely that lateral $CO_2$ flux occurs within the snowpacks at our field sites, especially with the presence of wind slabs, sun crusts, and ice lenses at the sites. These features are unaccounted for in our modelling, as modelling lateral $CO_2$ transport through a snowpack in addition to vertical transport would require a 3-D model. Our overall objective with this model was to observe and understand the differences in diffusive and advective transport through snowpacks. As such, we refrained from overcomplicating the 1-D model (e.g. Fick's second law of diffusion).

For sensitivity analysis, we calculated fractional change. Each post-wind event $CO_2$ value was compared to a $CO_2$ value under the same conditions as if a wind event had not occurred:

$$\text{fractional change} = \left| \frac{w-n}{n} \right|,$$

where $w$ is a post-wind event and $n$ is an event under no elevated wind conditions.

## 2.4 Field-model comparisons

In order to properly compare the field and modelled data, we determined the rate at which modelled $CO_2$ responded to the induced "wind events". This refers to the change in $CO_2$ concentration over time (ppm s$^{-1}$) as a result of the change in snowpack $CO_2$ diffusivity after initialization. Of the modelled data, we considered only scenarios with a soil diffusivity of $1.00 \times 10^{-7}$ m$^2$s$^{-1}$. Additionally, only "low wind" and "high wind" events were considered, which had induced snow diffusivities of $1.72 \times 10^{-5}$ m$^2$s$^{-1}$ and $9.08 \times 10^{-5}$ m$^2$s$^{-1}$, respectively. Output included $CO_2$ concentration at every 10 cm within the modelled environment (both soil and snow). For field-model comparison purposes, we only considered the $CO_2$ concentration of the topmost layer of snow.

We processed the enhanced concentration profile experiment data (winter 2015: 15 April–29 April) by calculating the rate of change of $CO_2$ concentration (ppm) per unit time (s) after a noticeable wind event.

## 3 Results

### 3.1 Snowpack $CO_2$ concentration profile experiment

Initial field campaigns (2014) showed a relationship between wind speed and $CO_2$ concentration within the snowpack at NM1 and NM2. Wind speed sometimes had a very strong effect on $CO_2$ concentration within the snowpack (Figs. 3 and 4).

Trace amounts of snow at NM1 and NM2 began accumulating at the beginning of data collection (11 November 2013), with appreciable ($> 25$ cm) snowfall at both stations occurring on 15 December 2013, and remaining through the winter. Maximum snow depth at NM1 was 188 cm (26 March 2014), whereas maximum snow depth at NM2 was 137 cm (4 January 2014).

There was a negative correlation between average wind speed and $CO_2$ concentration 50 cm above the ground, an example of which can be seen in Figure 3a. During this period of 31.5 h, snowpack $CO_2$ concentration at this height above soil ranged from 587 ppm to 965 ppm. Wind speeds over this same time period ranged from 3.2 km h$^{-1}$ to 31.1 km h$^{-1}$. The corresponding linear regression (Fig. 3b) shows the effect that average wind speed exerted on $CO_2$ concentration (R$^2 = 0.70$, $P < 0.001$). As wind speed increased, $CO_2$ concentration decreased at a rate of 14.4 ppm km$^{-1}$h.

Figure 4 shows measurements at NM1 over the same time period from 125 cm above ground. These $CO_2$ values were very close to predicted atmospheric concentrations, as the average snow depth over this time period at NM1 was 124 cm, very near the measurement height. The closeness of the measurement height to the snow surface indicates these values were likely a good representation of the $CO_2$ concentration at the snow-air interface. Despite increased atmospheric mixing, average wind speed

exerted good control over $CO_2$ concentration (Fig. 4a). This result is reinforced with the corresponding linear regression (Fig. 4b; $R^2 = 0.53$, $P < 0.001$). As wind speed increased, $CO_2$ concentration decreased at a rate of 1.57 ppm km$^{-1}$h.

We conducted a regression analysis of $CO_2$ concentration versus average wind speed for filtered data for winter 2014 (11 November 2013 to 26 March 2014), as per the three conditions specified in the Methods section. From this summary table (Table 2), there were some identifiable trends with the increasing height of $CO_2$ concentration measurement. With the increase from 50 cm to 125 cm at NM1 and 0 cm to 125 cm at NM2, there was a decrease in the y-intercept, which was the mean predicted value of $CO_2$ concentration if average wind speed was 0 km h$^{-1}$. Additionally, the average slope of individual regressions became flatter with an increase in measurement height. Finally, the strength of the relationship ($R^2$) decreased with an increase in measurement height (towards the open air). Instrumentation error for the NM1 0 cm $CO_2$ probe prevented data collection at that height.

The measurements that satisfied all conditions accounted for an average of 15.1% of the data collected at a given station (NM1, NM2) and height in the snowpack (0, 50, 125 cm). This value does not represent an estimate of the $CO_2$ flux during the snow-covered period, since we used a biased filtering process to identify wind events during periods of steady snow cover.

### 3.2   Enhanced concentration profile experiment

We collected $CO_2$ concentration profile data at the enhanced NM2 station from 16:00 on 15 April 2015 to 11:00 on 29 April 2015, a total of 331 uninterrupted hours (Fig. 5). Average snow depth over this time period was 157 cm, ranging from 149 cm to 167 cm. Average air temperature was $-1.4°C$, ranging from $-8.6°C$ to $7.6°C$.

Figure 5 shows a time series of $CO_2$ concentration throughout the snowpack (0, 25, 50, 75, and 100 cm from the ground), atmospheric $CO_2$ concentration (250 cm from the ground), and mean wind speed. There was considerable variability in snow-pack $CO_2$ concentration and wind speed over the two week period, with snowpack $CO_2$ values ranging from 357 ppm to 4161 ppm and wind speeds ranging from 0.0 km h$^{-1}$ to 34.0 km h$^{-1}$. Average wind speed over the two week period was 13.5 km h$^{-1}$.

Average $CO_2$ concentration decreased with increasing proximity to the atmosphere: 1244, 1076, 1007, 886, and 867 ppm at 0, 25, 50, 75, and 100 cm, respectively. Average atmospheric $CO_2$ concentration over this sampling period was relatively constant at 512 ppm. For some time periods between 15 April and 29 April 2015, there was a slight negative correlation between wind speed and snowpack $CO_2$ concentration (Fig. 5), however, this was not tested using the methodology of testing the winter 2014 data.

### 3.3   Modelling

Figure 6 shows results from sensitivity testing of an enhanced diffusion model used to simulate advection, and the effect of several parameters as deviations from a base case (Table 1). Model activity was investigated at the following layers: the topmost snow layer ($CO_2$ concentration in Fig. 6a and storage flux out of the top of the layer in Fig. 6c), the bottommost snow layer ($CO_2$ concentration in Fig. 6b), and the topmost soil layer ($CO_2$ concentration in Fig. 6d).

Results are shown as fractional depletion of $CO_2$ concentration in the snowpack (Figs. 6a, 6b, 6d), and factor increase in short-term $CO_2$ storage flux (Fig. 6c). Of the three parameters (soil diffusivity, snow diffusivity mimicking advection, and snow depth), soil diffusivity had negligible control on layers involving snow (Figs. 6a, 6b, and 6c), and is therefore not represented in those panels. Soil diffusivity showed some control on the modelled soil layer (Fig. 6d).

We also considered time when analyzing the modelled data to investigate how $CO_2$ concentration is affected during the "wind event" recovery period as the system works its way towards equilibrium (immediate change) and once the modelled system had recovered to an equilibrium state. Equilibrium refers to no change in the modelled storage flux, or when storage flux had returned to the initialized condition of 1 $\mu$mol m$^{-2}$s$^{-1}$. The two time "scenarios" considered were: 1) at 10 minutes, and 2) at 8 days following the simulated "wind event". The 10 minute scenario represented "immediately following a wind event" and the 8 day scenario represented "once equilibrium had been reached".

In the modelled topmost layer of snow (Fig. 6a), the maximum fraction to which $CO_2$ concentration was depleted was 0.39, once equilibrium was reached after a severe wind event. Snow depth had no effect on $CO_2$ depletion for either equilibrium scenarios at the top of the snowpack. For scenarios immediately following a wind event, severe winds had a greater effect on the fraction of $CO_2$ depleted, but this effect decreased with increasing snow depth (approaching no $CO_2$ depletion).

$CO_2$ concentration at the bottommost layer of snow (Fig. 6b) behaved similarly to the $CO_2$ concentration in the topmost layer. Depletions at the bottom of the snowpack were up to two times that of the depletions at the top of the snowpack (maximum fraction of 0.81 once equilibrium was reached after a severe wind event, with 100 cm of snow). Scenarios that immediately followed a wind event showed that severe winds had a greater effect on $CO_2$ depletion, although this decreased with increasing snow depth, reaching a minimum fraction of 0.06 at 100 cm.

Storage flux from the top of the snowpack into the modelled atmosphere is shown as a factor increase in short-term $CO_2$ flux (Fig. 6c). Scenarios at equilibrium (at 8 days post-event) are not shown, as there was no change in $CO_2$ concentration once equilibrium was reached. Of the scenarios that immediately follow a wind event, light and severe winds had similar effects on factor increase with 20 cm of snow: a factor of 0.53 (light wind) and a factor of 0.25 (severe wind). With increasing snow depth, severe winds showed a much greater fractional increase (9.92) in storage flux than light winds (1.15).

At the topmost soil layer (Fig. 6d), $CO_2$ concentration was affected by soil diffusivity and unaffected by snow depth. With increasing soil diffusivity at equilibrium, a greater fraction of $CO_2$ was depleted from the soil layer. Severe winds depleted a greater fraction than light winds. There was essentially no effect on the fraction of $CO_2$ depletion immediately following wind events (at 10 minutes post-event) of any severity, and therefore there is significant overlap of the two 10 minute lines in Fig. 6d.

## 4 Discussion

### 4.1 Wind causes short-lived advective anomalies

Findings of the initial snowpack $CO_2$ concentration profile experiment showed that there was a negative correlation between wind (advective) events and the $CO_2$ concentration in a snowpack, on a timescale of hours to days. This was clear from

specific examples (Figs. 3 and 4), as well as from the overall summary of linear regressions performed between $CO_2$ snowpack concentration and wind speed (Table 2). However, this was not continuous over the entire winter and was only true under particular conditions where filtering criteria were satisfied. The balance of the datasets that did not meet criteria were simply noisy with visible but weak trends. These time periods that did not meet the criteria may have resulted from the presence of vertical density variations (wind slabs, ice lenses) within the snowpacks at our field sites, plausibly causing lateral $CO_2$ flux. In addition to finding a negative correlation between wind events and $CO_2$ concentration within the snowpacks, analysis of data from the first experiment showed that there was a $CO_2$ concentration gradient throughout the snowpack, with highest concentrations closest to the soil and lowest concentrations closest to the atmosphere. This was consistent with previous literature, which indicates that the closer in the porous medium to the source of production of the trace gas (e.g. $CO_2$), the higher the concentration (Seok et al., 2009).

This work reinforced earlier observations of depleted $CO_2$ concentrations in field datasets (Seok et al., 2009), although we did not measure or calculate $CO_2$ storage flux directly in the field at the snow surface. However, we inferred that sporadic changes in snow-atmospheric flux would have been present from the large decreases in concentration. Positive storage fluxes were balanced by negative storage fluxes following wind events. It is important to consider concentration gradients to help with our understanding of the underlying physical processes of $CO_2$ transport through snowpacks.

As the measurements taken at each of the snowpack heights at each of the stations satisfied all specific conditions for an average of 15.1% of the time analyzed, we can conclude that advection showed some control over snow $CO_2$ transport for this location for the equivalent of 20.4 days during the 135 day period in 2014 (12 November 2013 to 26 March 2014). This value did not represent the percentage of annual flux during the snow-covered season (Liptzin et al., 2009), though did confirm that advective transport needed to be taken into account when studying snowpack $CO_2$ transport. It also gives an indication of how much data was eliminated for analysis, biasing our results.

The enhanced concentration profile experimental data reinforced the results of the initial findings and added $CO_2$ concentration measurements throughout the snowpack, increasing the total in-snow measurements from three to five. This gave us a clearer indication of how the $CO_2$ concentration gradient behaved, even without taking snow properties into account. This data covered the late winter period, so ice layers within the snowpack were likely present. Despite this, the wind seemed to have an effect on $CO_2$ snowpack concentrations, even at 0 cm with a snowpack of 157 cm.

Some authors have used turbulent atmospheric pressure pumping to explain anomalous $CO_2$ storage fluxes, but have often focused this work on shorter, high frequency timescales of seconds to minutes (Massman et al., 1995). On the longer, low frequency range of the timescale, Bowling and Massman (2011) and Massman et al. (1995) mentioned the importance of synoptic scale changes in atmospheric pressure. Additionally, Rains et al. (2016) showed that changes in wind speed at multiple hour frequencies (greater than 10 hours) could be more effective than atmospheric pressure pumping when explaining changes in snowpack $CO_2$ concentration. These processes of different timescales and different mechanisms (atmospheric pressure changes, wind) affect $CO_2$ concentration gradients and fluxes measured with Fick's law by ventilating diffusive media, like snowpacks. The ventilation, no matter the timescale, affects the $CO_2$ concentration gradient by mixing atmospheric air into diffusive media where $CO_2$ typically pools, thereby affecting the $CO_2$ flux from the top of the snowpack. Our work showed

how persistent wind and an enhanced diffusive profile controlled $CO_2$ concentration and fluxes across timescales of hours to days, in the midrange between very high frequency pressure pumping and low frequency barometric pressure effects. The low frequency, synoptic processes occur on a longer time scale than the wind depletion events discussed in this study, though would be present here as well, and would likely contribute to some of the variability (Robinson and Sextro, 1997; Tsang and Narasimhan, 1992). With more longer continuous wintertime $CO_2$ records, similar to this one, it may be possible to extricate these synoptic process periodicities in addition to the mid-range frequencies we investigated.

## 4.2 A diffusive model can help explain advective questions

The 1-D diffusional transport model and enhanced diffusion approach was able to replicate the $CO_2$ depletions seen in the field in this experiment, as well as those in previous observations (Seok et al., 2009) and in other plausible situations. Advective events were created with induced increases in snowpack diffusivity after model initialization, which worked well to mimic wind events.

In general, when snowpack diffusivity was instantaneously increased in this diffusive transport model, we observed rapid changes in the snowpack $CO_2$ concentration, $CO_2$ storage flux, and soil $CO_2$ concentration, similar findings to Bowling and Massman (2011). This effectively simulated advective events observed in the field. According to this model, severe wind events always produced more dramatic results than light wind events in terms of both rate of change (flux) and overall concentration change.

This modelling work showed that we can simplify the impacts of sustained advection on $CO_2$ in a soil-snow system to an effective diffusion problem. This approach was less complicated than other models that use the diffusive-advective coupled solution approach.

## 4.3 Field-model comparisons

To determine the applicability of the model to real-world scenarios, we compared our field and model results. To do so, we calculated the rate of change of $CO_2$ concentration (ppm) per unit time (s) after a wind event for both the modelled wind events and the field wind events (using the 2015, enhanced experiment). Figure 5, which displays a time series of $CO_2$ concentrations and wind speed over two weeks in late April 2015, shows that despite similarly variable wind conditions, snowpack $CO_2$ concentrations throughout the first week vary less than the $CO_2$ concentrations observed in the second week. The lack of variation in the first week could be due to a variety of reasons, including the composition of the snowpack, or other meteorological conditions like temperature or humidity. Despite the variation through the two week period, it was still possible to discern change in $CO_2$ concentration after a wind event (Table 4).

Table 3 summarizes the calculated rates of change of modelled $CO_2$ concentration at varying snow depths, at low and high simulated wind speeds (induced change in snow diffusivity), and at various times since the modelled wind event. All of these modelled measurements were taken from the topmost snow layer. Table 4 shows a similar summary for four wind events in the field in April 2015. All of these $CO_2$ field measurements were taken at 100 cm from the ground within the snowpack, which was the in-snow measurement farthest from the ground and closest to the atmosphere at the time.

Change in modelled $CO_2$ concentration per second (Table 3) did not align perfectly with the change in field $CO_2$ concentration per second (Table 4) after a wind event. However, the rates of change in the field events ($-0.07, -0.04, -0.20. -0.04$ ppm s$^{-1}$) were of approximately the same order of magnitude as the rates of change in the modelled events (ranging from 0.00 to $-2.08$ ppm s$^{-1}$). This indicated that the model was able to mimic advective events with some accuracy. Though it may be possible like in other studies (Latimer and Risk, 2016) to apply an iterative procedure to our model with the conditions we observed in the field (e.g. initial $CO_2$ concentration), we deemed that to be unnecessary. This is because our primary goal was to properly illustrate the underlying physics of $CO_2$ transport through snowpacks. As such, matching the model conditions exactly to the field conditions was not required.

This study showed the importance of continuous monitoring of $CO_2$ concentrations and fluxes from soils through snowpacks. Similarly, Webb et al. (2016) and Rains et al. (2016) highlighted the non-growing season contributions to annual $CO_2$ flux. Webb et al. (2016) showed that different wintertime measurement methods at one Alaskan site resulted in a fourfold range in $CO_2$ loss. The eddy covariance (EC) method showed the highest fluxes, as more $CO_2$ was released under windy conditions and the EC method was able to measure fluxes in turbulent conditions (Webb et al., 2016). Rains et al. (2016) noted that there are benefits and disadvantages to the EC, flux gradient, and chamber methodologies for non-growing season soil $CO_2$ flux measurements, and that accurate parameterization of advective transport through snowpacks is important, regardless of the methodology. This is particularly true because of the likelihood that $CO_2$ flux through snowpacks is often underestimated (Rains et al., 2016). Accompanying these findings, we agreed that infrequent measurement can lead to significant error in the annual C budget of various ecosystems once inaccurate values are scaled up (Fig. 7). The effects of advection on these soil-snow systems can lead to variability in storage flux, as effective diffusion is closely related to wind. Snowpack depth, density, and layering will also affect the timing and amounts of $CO_2$ storage flux from these systems. We recommend that future studies utilize continuous $CO_2$ monitoring methods and consider the advective effects of wind, in order to capture the uncertainties of soil $CO_2$ emissions in snow-covered ecosystems.

## 5  Conclusions

Although this study was conducted at one site over two winters, the findings have implications for measuring wintertime $CO_2$ fluxes in snow-covered environments. This is important for continued careful assessment of the soil C pool and fluxes of these snow-covered regions, which are experiencing increasing temperatures and variations in precipitation patterns.

As seen from the fieldwork in winters 2014 and 2015, advective transport by wind is important for $CO_2$ concentration (and therefore flux) through a soil-snow profile. Additionally, this process can be simulated with some accuracy by a model of enhanced diffusion. In both field and model cases, we observed how sustained winds could deplete $CO_2$ concentration in the snowpack, and create storage flux outward to the atmosphere. During the re-equilibration phase, fluxes across the snow-air interface would have been depressed, as most of the production contributed initially to pore space storage. This process of buildup and release occurs with regularity in snow profiles, and is likely more severe in snowpacks than in soil, which has lower permeability and is therefore less vulnerable to wind invasion.

Transport lags are the main effect of diffusion and advection. Measurements such as eddy covariance, which can be made above the snow profile with speed, are at an advantage for detecting storage flux events. While useful for total accounting purposes, eddy covariance records may not be effective in determining specific overwinter biological soil $CO_2$ production. For this, sensors within or at the base of the snowpack would also be needed, allowing the results to quantify soil-snow fluxes or concentration gradients within the first few centimetres of snow. Additionally, in situ sensors are typically cheaper and can be more easily and frequently deployed than eddy covariance methods. Alternatively, the model used here, which accurately simulated gas transport physics, could be applied through an inversion scheme to determine microbial changes in $CO_2$ production by removing the effects of snow gas transport.

This study shows snow profile $CO_2$ depletions that exist on timescales of hours to days. Putting this knowledge into practice would help to improve our understanding of global winter soil $CO_2$ release because it improves our efforts to quantify winter fluxes. As a start, we recommend that researchers approach winter data like they do summer data, which means that they should consider using continuous automated approaches for wintertime $CO_2$ flux observations, as done in this study. We also recommend close collaboration between the modelling community and soil field scientists. This will ensure that available physical models are being effectively used for stripping flux data of transport-related artefacts, thereby isolating soil biological behaviour.

*Competing interests.*  Authors have no competing interests to declare.

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

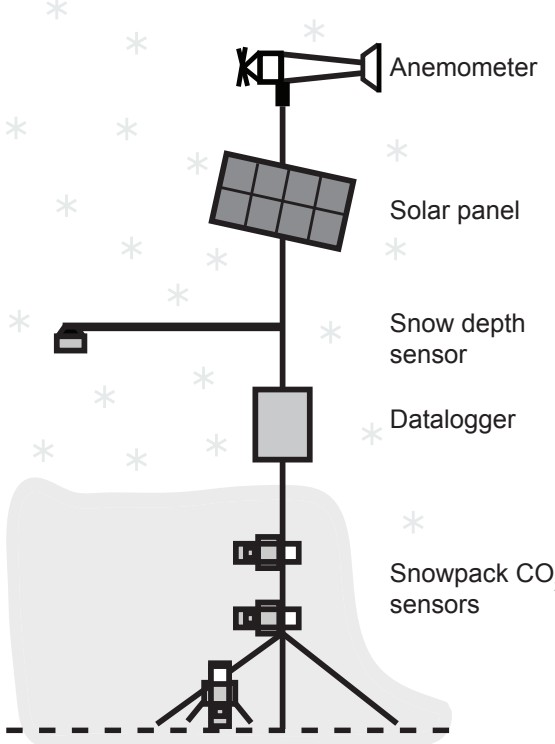

**Figure 1.** Schematic of initial (2014) $CO_2$ monitoring stations (NM1, NM2) at North Mountain, Cape Breton. Snowpack $CO_2$ sensors were at 0 cm, 50 cm, and 125 cm within the snowpack (diagram not to scale).

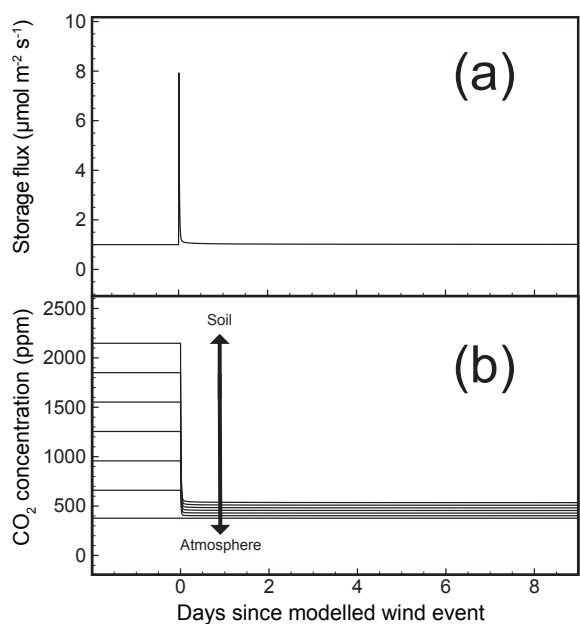

**Figure 2.** An instantaneous change in snowpack diffusivity after model initialization mimics advection. **(a)** Shows the modelled storage flux with an induced change in snowpack $CO_2$ diffusivity. **(b)** Shows the corresponding change in snowpack $CO_2$ concentration at every 10 cm. Soil diffusivity $= 1.00 \times 10^{-7}$ $m^2 s^{-1}$, stepped snow diffusivity $= 9.08 \times 10^{-5}$ $m^2 s^{-1}$, and snow depth $= 60$ cm. The Soil-Atmosphere arrow indicates depths within the 60 cm snowpack: highest modelled $CO_2$ concentrations occur at the soil-snow interface, whereas lowest modelled $CO_2$ concentrations occur at the snow-atmosphere interface, before and after the advective "wind event".

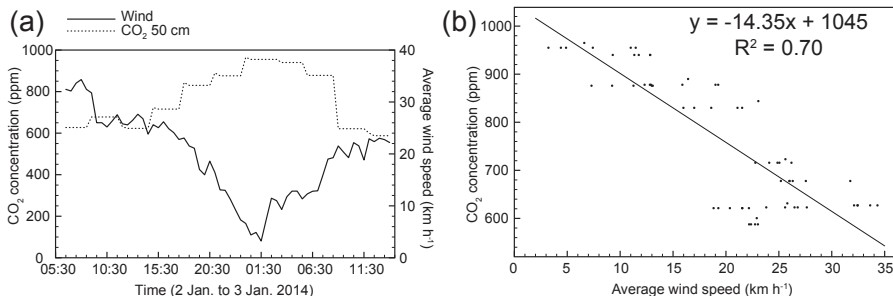

**Figure 3. (a)** Time series of wind speed and $CO_2$ concentration at 50 cm above the ground within the snowpack from 06:30 on 2 January 2014 to 14:00 on 3 January 2014 at NM1. Average snow depth at NM1 over this time period was 124 cm. **(b)** The corresponding linear regression of $CO_2$ concentration versus average wind speed ($R^2 = 0.70$, $P < 0.001$).

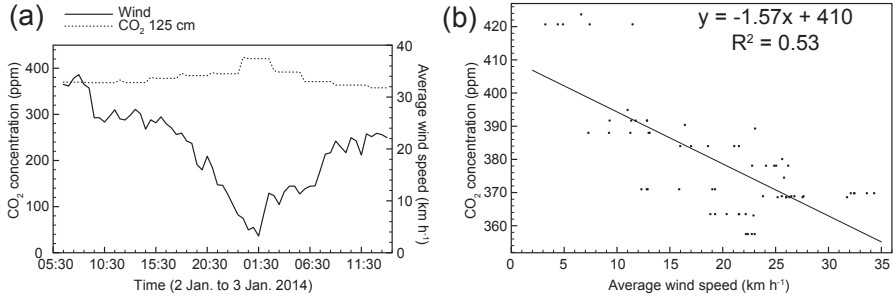

**Figure 4. (a)** Time series of wind speed and $CO_2$ concentration at 125 cm above the ground from 06:30 on 2 January 2014 to 14:00 on 3 January 2014 at NM1. Average snow depth at NM1 over this time period was 124 cm. Therefore, these $CO_2$ values were a good representation of the snow-air interface. **(b)** The corresponding linear regression of $CO_2$ concentration versus average wind speed ($R^2 = 0.53$, $P < 0.001$).

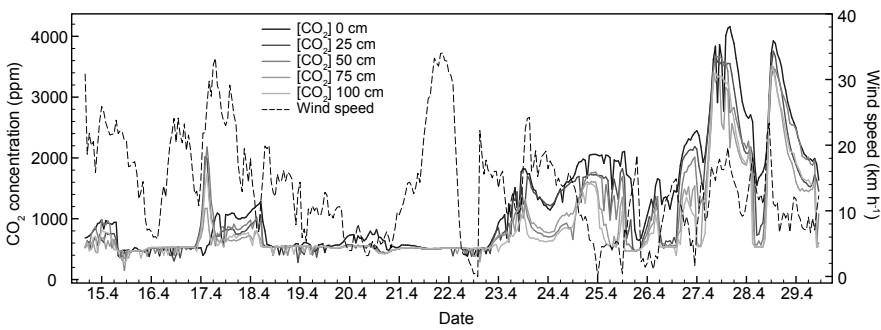

**Figure 5.** Time series of enhanced profiler experiment (winter 2015) $CO_2$ concentrations throughout the snowpack and wind speed at NM2 over 2 weeks during late winter 2015 (15 April–29 April). Measurements were recorded hourly.

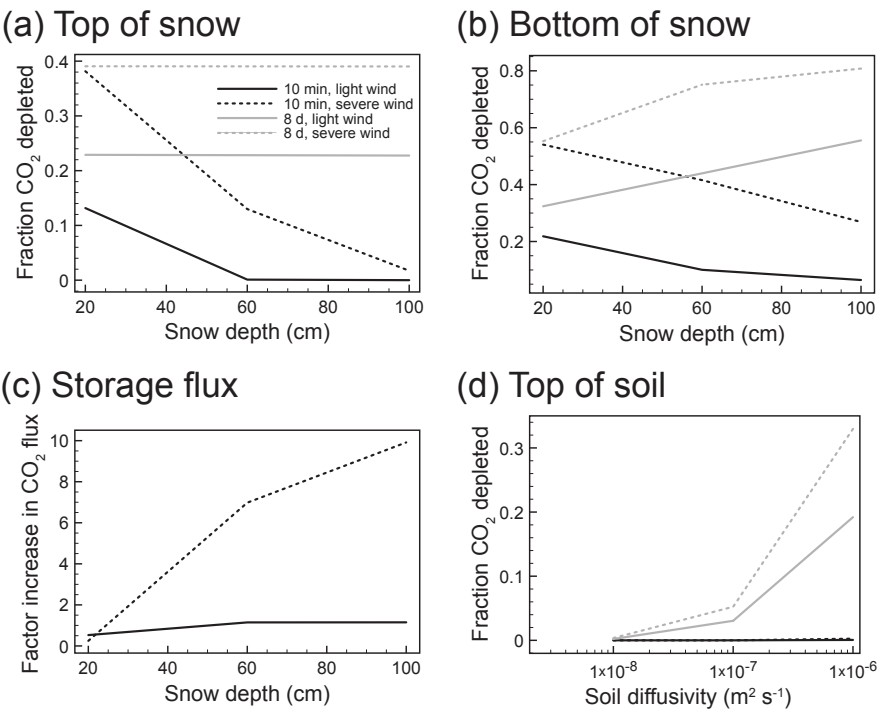

**Figure 6. (a)** Modelled results at top of snowpack shown as the fraction of $CO_2$ concentration depleted from the snowpack. **(b)** Modelled results at bottom of snowpack shown as the fraction of $CO_2$ concentration depleted from the snowpack. **(c)** Modelled storage flux, shown as factor increase in short-term $CO_2$ flux. Scenarios at equilibrium (8 days) were incalculable (not shown), as there was no change in $CO_2$ concentration once equilibrium was reached. **(d)** Modelled $CO_2$ at the topmost soil layer, shown as the fraction of $CO_2$ concentration depleted from the snowpack. There was very minimal effect on the fraction of $CO_2$ depletion for immediate scenarios (10 minutes), and so there is significant overlap of the two 10 minute scenarios (light wind and severe wind).

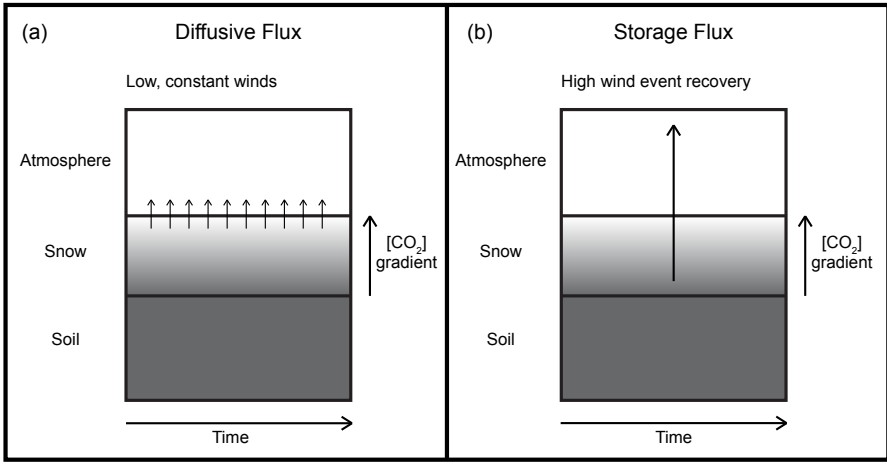

**Figure 7.** Conceptual diagram of diffusive versus storage flux. **(a)** Diffusive flux through a snowpack, with $CO_2$ originating from soils and consistently passing through a diffusive medium into the atmosphere as a result of a concentration gradient. Small arrows indicate low levels of diffusive flux that are prevalent and constant through time. **(b)** Storage flux through a snowpack, with $CO_2$ originating from soils, pooling in a diffusive medium, and then released to the atmosphere at a higher rate (than diffusive flux) following a high wind event, which has ventilated the top of the diffusive medium and steepened the concentration gradient. One, larger arrow indicates the higher rate and lower frequency of storage flux out of snowpacks when compared with diffusive flux.

**Table 1.** A 1-D soil $CO_2$ diffusion model was adapted for the soil-snow system. The model simulated step changes in transport rate over a broad range of plausible synthetic cases. Soil diffusivity ranged logarithmically, whereas snow diffusivity and snow depth ranged linearly. We ran the model with all possible permutations of these parameters.

| Parameter | Range of values | Number of values tested |
|---|---|---|
| Soil diffusivity | $1 \times 10^{-8}$ to $1 \times 10^{-6}$ $\mathrm{m^2 s^{-1}}$ | 3 |
| Snow diffusivity at step change | $8 \times 10^{-6}$ to $9.08 \times 10^{-5}$ $\mathrm{m^2 s^{-1}}$ | 10 |
| Snow depth | 20 cm to 100 cm | 3 |

**Table 2.** Summary of regression analysis between $CO_2$ concentration within the snowpack and wind speed. Data were filtered to satisfy the following conditions: 1) snow cover was considered to be at equilibrium, 2) the relationship produced a slope $< 0$, and 3) $R^2 \geq 0.1$. N is the number of time periods that satisfy all 3 conditions. Each time period covered a minimum of six hours. Y-intercept is the mean $CO_2$ concentration when wind speed $= 0$ km h$^{-1}$. Slope is the mean change in $CO_2$ concentration with a 1 km h$^{-1}$ increase in wind speed. $R^2$ is the mean strength of the relationship between $CO_2$ concentration in the snowpack and mean wind speed. n is the mean number of half-hourly observations within each N. Duration is the mean duration of N. Instrumentation error for the NM1 0 cm $CO_2$ probe prevented data collection at that height.

| Site | Snow depth | Height in snowpack | N | y-intercept | Slope | $R^2$ | n | Duration |
|------|-----------|--------------------|---|-------------|-------|-------|---|----------|
| | cm | cm | | ppm | ppm km$^{-1}$h | | | h |
| NM1 | $708 \pm 600$ | 0 | n/a | n/a | n/a | n/a | n/a | n/a |
| | | 50 | 29 | $1399.2 \pm 1000$ | $-23.2 \pm 30$ | $0.41 \pm 0.2$ | $30 \pm 20$ | $15 \pm 10$ |
| | | 125 | 27 | $642.3 \pm 700$ | $-12.0 \pm 30$ | $0.36 \pm 0.2$ | $29 \pm 20$ | $15 \pm 10$ |
| NM2 | $625 \pm 300$ | 0 | 29 | $1196.8 \pm 500$ | $-13.1 \pm 8$ | $0.49 \pm 0.2$ | $38 \pm 30$ | $19 \pm 20$ |
| | | 50 | 22 | $547.4 \pm 200$ | $-6.8 \pm 10$ | $0.35 \pm 0.2$ | $50 \pm 80$ | $25 \pm 40$ |
| | | 125 | 25 | $379.2 \pm 7$ | $-0.5 \pm 0.5$ | $0.29 \pm 0.2$ | $41 \pm 30$ | $21 \pm 20$ |

**Table 3.** Summary table of change in modelled $CO_2$ concentration per second at 1, 2, 4, 6, and 24 h since the wind event (step change in modelled snowpack diffusivity) at the topmost layer in the model. Snow depths of 20, 60, and 100 cm are shown, along with lowest and highest simulated wind speeds.

| | | Time since wind event (h) | | | | |
| --- | --- | --- | --- | --- | --- | --- |
| | | 1 | 2 | 4 | 6 | 24 |
| **Snow depth** | **Relative wind speed** | Rate of change of $CO_2$ | | | | |
| cm | | ppm s$^{-1}$ | | | | |
| 20 | low | −0.55 | −0.20 | −0.06 | −0.03 | 0.00 |
| 20 | high | −0.03 | −0.01 | −0.01 | 0.00 | 0.00 |
| 60 | low | −0.80 | −0.64 | −0.38 | −0.24 | −0.03 |
| 60 | high | −1.71 | −0.67 | −0.22 | −0.11 | −0.01 |
| 100 | low | −0.16 | −0.26 | −0.27 | −0.23 | −0.06 |
| 100 | high | −2.08 | −1.24 | −0.54 | −0.29 | −0.02 |

**Table 4.** Summary table of change in actual $CO_2$ concentration per second for four events in April 2015 when a decrease in $CO_2$ concentration corresponded to an increase in wind speed. $CO_2$ concentration was measured in the snowpack at 100 cm from the ground. Rate of change of $CO_2$ concentration, snow depth, start time, end time, range of $CO_2$, and range of wind speed are given in the table.

| Event number | 1 | 2 | 3 | 4 |
|---|---|---|---|---|
| Rate of change of $CO_2$ (ppm $s^{-1}$) | −0.07 | −0.04 | −0.20 | −0.04 |
| Snow depth (cm) | 162 | 152 | 155 | 156 |
| Duration of ppm decrease (h) | 4 | 3 | 2 | 14 |
| Initial $CO_2$ (ppm) | 1733 | 1105 | 2061 | 3445 |
| Final $CO_2$ (ppm) | 648 | 690 | 596 | 1771 |
| $CO_2$ decrease (ppm) | 1085 | 415 | 1465 | 1674 |
| Duration of wind increase (h) | 8 | 4 | 5 | 4 |
| Initial wind value (km $h^{-1}$) | 10.8 | 10.5 | 9.2 | 11.0 |
| Final wind value (km $h^{-1}$) | 33.2 | 24.2 | 18.1 | 23.4 |
| Wind increase (km $h^{-1}$) | 22.4 | 13.8 | 8.9 | 12.3 |