# Peer review of "Explaining CO2 fluctuations observed in snowpacks"

_Biogeosciences, 2017_

## Referee Comment (RC1) · F. A. Rains (Referee) · 25 Jul 2017

General Comments

First of all, I would like to say this was a very well put together study. Having performed wintertime respiration measurements myself, I know that it is not an easy task, kudos. Also, the system design seems robust, and accurate. Please take the below questions/comments with an open mind. Does the rate of flux affect the total quantity of CO2 released to the atmosphere? A bit of a rhetorical question, however this seems pertinent. Its clear that total C released is obviously a significant metric, but perhaps you could expand on how/why the rate of release is significant.

You mention a trend of thinning snowpack in North America over the last number of

decades. Also mentioned is the insulating effect that a deeper snowpack plays in allowing microbes to exist/or allowing for microbial respiration. It would follow logically then that assuming no change in air temperature, a thinning snowpack would decrease microbial, wintertime respiration by allowing the soil to reach sub 0 Celsius, or whatever that lower threshold may be. This may be slightly off topic but it seems related and pertinent. This could perhaps be addressed by mentioning other long term meteorological trends in North American winters, such as average air temperature, etc. . .

Content Comments

Section 1, lines 28-29. An example of "underestimating" winter contribution to atmospheric C would be supportive of your statement. It seems that assumptions are being made that current models assume that the wintertime contribution is nil. In fact some models may over estimate this variable. Again, an example of a widely used, modern model that excludes or under represents wintertime production of CO2 would be illustrative.

Section 2.3 Model Development. Line 30. How did you calculate snow pack porosity, and tortuosity? Was snow pack density measured at different intervals or assumed homogeneous for the different "steps"? Also, Fick's 1st Law of Diffusion is adequate for explaining flux in a 1-dimensional, relatively homogeneous medium. However we know that a snowpack stratigraphy is highly variable in space and time. Furthermore, assuming the non-static/non-homogeneous nature of wind and how it affects the snowpack in a very localized manor, could lateral flux occur with the snow pack. Also, elaboration on the role of dense wind slabs, sun crusts, and other ice crusts/lenses within the snowpack would be enlightening. In addition, it seems plausible that Fick's 2nd Law of Diffusion could potentially be useful.

Conclusion. Why is total "accounting" via eddy covariance lacking in this regard? At the outset it would appear that eddy covariance can tell you not only the rate of flux, but the net production of CO2 for a given footprint (accounting?), while eliminating margin

for error i.e. snowpack variability. What other sources of $CO_2$ would be accounted for in addition to soil respiration that would not allow you assume all measured net wintertime $CO_2$ was in fact from the soil? An few more sentences explaining your statements/reasoning that in-situ $CO_2$ probes are superior would be enlightening.

Technical comments

Line numbering appears off, continues from abstract through first portion of introduction, and then switches back mid way. No other technical or grammatical errors were noted.

---

## Referee Comment (RC2) · Anonymous Referee #2 · 28 Jul 2017

This work presents both field data and a simple model to address a methodological problem with winter CO2 flux measurements. While both field and model data are presented it isn't really a model-data comparison as the field data isn't directly compared to the model data. As currently structured, there isn't a convincing narrative nor is it clear what is novel. While continuous CO2 datasets are not that abundant, the authors don't present that data and focus on a confusing method of comparing CO2 concentrations to wind speeds. They have adapted a soil diffusion model to the soil-snow system. I'm not sure if the model is too simplistic or if the text just needs greater clarity, but I couldn't follow how this model could help explain the field data. Figure 2 shows that it can create a step change in CO2 concentration gradient, but that isn't enough to be able to match the field data. The paragraph starting on P10 L8 describes how

combining modeling and field data could be really powerful, but the temporal changes in the CO2 concentration gradient related to advection that we know happen based on the field data presented here and elsewhere weren't clearly shown. The writing is generally good; my main criticism is that the flow of the narrative, in particular the connections between paragraphs, could be improved. In addition, both the introduction and conclusion sections contain overly broad statements that aren't supported by the rest of the paper. There is certainly need for this type of work and perhaps with some modifications to the model and/or greater clarity of what was done in both the text and the figures could show how the model and field data can be compared, this paper would be acceptable for publication. Finally, I appreciate the authors presenting all the CO2 data. However, it is somewhat surprising to me that snowpack CO2 concentrations could be as low as 151 ppm on Page 6 or well under 400 in Figure 4 or that the atmospheric concentrations was 512 ppm on Page 7, about 100 ppm higher than what it should be. Could the authors justify these seemingly strange measurements?

P1 L19-21 These lines are way too general for the rest of the paper. The paper is about making the winter flux techniques better not about the soil C and the global C cycle. Or at least it needs to be demonstrated how the results might directly affect the global C cycle.

P2 L1-3 The authors haven't presented any evidence for yet about why rates might be underestimated

P2 L4-5 This paragraph is just about using the diffusion gradient in the snowpack method to measure soil flux. This method should be explained and its advantages and disadvantages to the other methods should be described.

P4 L10-12 This is the key piece of knowledge that this paper is attempting to explain. There is a methodology for measuring CO2 flux that has known limitations. The net result is that it is difficult to separate variability from advection (an artifact) from microbial processes (the actual goal). It would be very helpful if there was a standard correction

that could be used with the diffusion based method to account for advection.

P2 L13-31 Somewhere in this paragraph it needs to be made clear that the assumption is that $CO_2$ production is happening in snow-covered soils, but there are methodological limitations to how the production is quantified. This paper is not about the controls per se, but about how to overcome the methodological limitations so that the mechanisms of $CO_2$ production can be studied.

P2 L32-34 It would be helpful to talk about of the role of timescale in advective processes in the introduction to justify why you are looking at the hours to days timescale. There is a mention of it in the discussion, but is important here too.

P3 L14 Delete the sentence starting with obviously. How do "variable meteorological conditions" affect snow depth?

P3 L16 How long were the measurements made during winter 2014? (as well as winter 2015 in line 34).

P3 L32-37 I'm confused by this paragraph. In the preceding paragragph, it says that at each of the two stations there were $CO_2$ measurements at 0, 50, and 125 cm, but now it says 25, 50, 75, and 100 cm for NM2. Similarly, in this paragraph, it says the data were collected hourly whereas above it was half hourly. Were the Eosense sensor in addition to Vaisala sensors at the depths above? These sensors need more description. You already said that snow depth and wind speed were measured at both stations in lines 29-31. Somewhere in the methods or results can you indicate what the timing of snow-cover and the maximum depth were.

P4 L5-7 You need to say why it is important for steady state conditions that snow depth had not changed. Is there a quantitative way that this was determined?

P4 L9-11 What does "ideal" mean?

P4 L18-22 I'm confused about the time period selection. Let's say the snowpack didn't change depth from January 1 to January 5th. Did you look at that whole time period
with one regression? In the caption for figure 2 it says that the average number of 30 minute measurements in the filtered dataset was 29-50 at different heights/sensors. Does that mean the snow depth was changing every day? Or was there some other criteria besides snow depth changing that set the length of time. It seems like the time period selection was based on finding a change in $CO_2$ after an increase in wind speed. Is this assuming that advection is equally affecting the snowpack from the soil surface to the snow surface? Is that a valid assumption? I'm not convinced that this is the right approach to determine the effects of wind on snowpack $CO_2$, but there needs to be a better justification and description of the approach.

Is there a different way to do this calculation with fewer assumptions? For example, could you calculate the $R^2$ for the snowpack concentration gradient every 30 minutes and plot that vs. wind speed as a test of whether the wind speed affects the predicted gradient with zero wind? Or compare the concentration gradient to the wind like Seok et al. (2009)? If you examine every time point there is the problem that the previous time points likely affect the relationships. Perhaps you could also try averaging different lengths of time (e.g. 30 minutes to 12 hours or longer)?

P5 L6-11 This paragraph needs to be clarified to describe what the model is doing. How are the initial conditions set What are "step changes in transport rate?" I thought there were step changes in the parameters. Based on figure 2 there is some constant $CO_2$ gradient before time zero and then there is a step change in the diffusivity and the $CO_2$ gradient adjusts quickly. Is there a fixed emission of $CO_2$ from the soil and then the combinations of the diffusion/depth parameters determine the $CO_2$ concentration gradient? Is there a temporary step change and then it returns to the initial value? Or does Storage flux needs to be defined, perhaps even in the introduction. Why is the soil diffusivity included in the model? How is snow depth included? Is the model run with all possible permutations of the 3 parameters?

P5 L24-25 What does "the rate at which modelled $CO_2$ responded to an induced wind event" mean? What is the rate? Is an induced wind event the same as an advective

wind event in line 15?

P5 L30-31 What is the "enhanced concentration profile experiment?" How were the data processed?

P6 L14-19. I'm confused by this example. I thought the ideal situation was when wind speed increased gradually and then abated. These figures show the opposite with wind speeds gradually decreasing and then increasing. Figure 3 seems like a good example, but I'm confused by Figure 4. The snow-atmosphere interface seems hard to measure. Is there a time period with a deeper snowpack that could be shown instead? Why are the $CO_2$ concentrations so much lower (360-390 ppm)than atmospheric (512 ppm)? There was a brief period when the concentration was around 420 ppm that seems to be driving the relationship in this case. If those few hours of data were removed, it doesn't look like there would be much of a relationship.

P6 L20-22 Wasn't data collected the whole winter? The data shown in figure 3 and 4 are not included in this time period. This should be clarified in the methods section.

P6 L22 How come there is no data for the 0 depth on NM1?

P6 L34 How can you get a concentration of 151 ppm $CO_2$ in the snowpack? Is $CO_2$ being consumed or is it some kind of measurement error? Similarly, why is the atmospheric $CO_2$ 512 ppm (P7 L5)? Is this a calibration error.

P7 L5-7 Either test for a relationship between wind speed and $CO_2$ or delete this sentence. It is surprising to me that so much of the first week or so there is no concentration gradient in the snowpack. That is on the 16th-17th and 19th-23rd the whole snowpack is essentially the same as the atmosphere. This seems somewhat surprising. It doesn't seem that much windier than the second week of the experiment. What is happening?

P7 L18, L23 What is equilibrium mean? Based on Figure 2 it seems like the model has some step change in parameter and then the $CO_2$ gradient adjusts instantaneously. Similarly, I don't know what a scenario is.

P7 L31-34 I'm not sure why soil diffusivity was included the model as a parameter.

P8 L6-13 There are two different phenomena described in this paragraph. One is that there is a monotonic decrease in CO2 from the soil surface to the snow surface if the only source of CO2 production is the soil. I'm sure with ice lenses or if the density of the snowpack isn't constant that there are ways this couldn't be true, but it seems like this should generally be true. The more important question is whether there is a relationship between CO2 concentration and wind speed. The authors have chosen to look over time to see if

P8 L14-17 You should either calculate storage fluxes or remove this paragraph.

P8 L18-26 I'm not sure why you can conclude that advective transport needs to be taken into account by the fact that during 33.6% of the time analyzed there is a re-lationship between CO2 and wind speed. These two paragraphs don't have much quantitative analysis in them.

P8 L27-35 Can you give a clearer description of how these processes that occur at different time scales would affect the CO2 concentration gradients and the fluxes mea-sured with Fick's Law? What are "a continuously enhanced friction velocity" and "an enhanced diffusive regime?" It seems like 54 days of measurements should be enough to capture some synoptic variability if you looked for it.

P9 L9-12 I'm not sure I understand exactly what the model did or what equilibrium means. If I look at figure 2 it seems like an instantaneous change in diffusion led to an instantaneous chance in concentration gradient. I don't see any change over time in CO2 concentration which is what I would have thought disequilibrium would be.

P9 L13-14 I don't understand these sentences.

P9 L16-27 Why not try to match the model conditions exactly to the field conditions to start at least in terms of CO2 concentration

P10 L6-7 Just like the beginning of the introduction, this sentence seems like an overreach with no connection to the rest of the text.

P10 L11-14 This seems like where model data synthesis could really move this field forward.

P10 L-15-21 Alternative measures of CO2 flux need to be discussed earlier in the manuscript.

P10 L22-27 This study show snow profile depletions, but I wouldn't say that it explains them. While I agree with the sentiments in the rest of the paragraph, they aren't direct conclusions of the work here.

Figure 1. This figure can be deleted. Or it needs to be improved so that the labels match up to the icons and the depths are shown.

Figure 2. Indicate that an instantaneous change in the diffusivity mimics advection. Storage flux needs to be defined in the text somewhere. Either call it storage flux or apparent storage flux. Indicate the depths on panel b.

Figure 3. Use the data not the record number on the x axis. Would you expect there to be a hysteresis because of advection? That is, if you drew a line connecting all these points in time would the concentration be higher than average when the winds are decreasing and then lower than average as winds are increasing again? Or maybe vice versa? I realize it is not a crucial question for the model-data comparison, but it seems important for converting concentration gradients into fluxes.

Figure 4. Why not pick a time to show when this sensor is really in the snowpack?

Figure 5. The different colors/dashes are hard to see. It would be better if the wind speeds were in a separate panel. Atmospheric CO2 probably isn't necessary to snow either.

Figure 6. The dashes are hard to distinguish. Why are there 4 cases for a and b but only 2 or 3 for c and d? Are the lines on top of each other? If so, make this clear in the

caption.

Figure 7. I like the idea of a conceptual figure, but I'm not sure why lots of little arrows represent diffusion and one big arrow represents advection. Can you make something that shows how the concentration gradient and fluxes change over time in response to wind? Maybe some combination of the information in Figure 2 and Table 3 along with a calculation of the flux using Fick's law and a calculation of the storage flux over time?

Table 2 is a good summary, but it seems like you can get rid of n as duration is essentially n/2

Table 4 Can these events also be shown on figure 5? The measurement depth is in the caption and can be removed from the table.

---

## Author Comment (AC1) · 8 Sep 2017

**Author response to comment on "Explaining CO$_2$ fluctuations observed in snowpacks" by Laura Graham and Dave Risk**

**By F.A. Rains (Referee 1)**

**(referee comments in black, author responses in blue)**

General Comments

First of all, I would like to say this was a very well put together study. Having performed wintertime respiration measurements myself, I know that it is not an easy task, kudos. Also, the system design seems robust, and accurate. Please take the below questions/comments with an open mind. Does the rate of flux affect the total quantity of CO2 released to the atmosphere? A bit of a rhetorical question, however this seems pertinent. It's clear that total C released is obviously a significant metric, but perhaps you could expand on how/why the rate of release is significant.

Thank you kindly. The referee's insightful and thoughtful comments will help clarify some key concepts throughout our study.

The question of whether the rate of flux affect the total quantity of CO2 released to the atmosphere is indeed important, and should not be overlooked. Though we mentioned in the Introduction the importance of organic C reserves in high latitude soils, we acknowledge that we neglected to make the connection to fluxes from soils and their contribution to atmospheric CO2 concentrations. This detail is added to the introduction, as suggested, along with corresponding references (e.g., Raich et al., 2002).

You mention a trend of thinning snowpack in North America over the last number of decades. Also mentioned is the insulating effect that a deeper snowpack plays in allowing microbes to exist/or allowing for microbial respiration. It would follow logically then that assuming no change in air temperature, a thinning snowpack would decrease microbial, wintertime respiration by allowing the soil to reach sub 0 Celsius, or whatever that lower threshold may be. This may be slightly off topic but it seems related and pertinent. This could perhaps be addressed by mentioning other long term meteorological trends in North American winters, such as average air temperature, etc…

Thank you for bringing this up. There are certainly some perceptible flaws in this logic, which can be clarified with the addition of detail regarding long-term meteorological trends in North American winters. One helpful assumption is that this thinning snowpack is a result of increasing air temperatures with the onset of anthropological climate change (rather than assuming no change in air temperature). Dyer and Mote (2006) help to address this, with their study indicating earlier onset of spring melt (associated with higher temperatures and variations in precipitation). The most important details are perhaps that there is still significant global snow coverage despite increasing global temperatures, and that soil respiration occurs beneath this snow. Thinning snowpacks are certainly prevalent on average, but air with the coldest temperatures have the lowest ability to hold water vapour—so more intense individual snow events are likely to occur with increasing air temperatures.

Content Comments

Section 1, lines 28-29. An example of "underestimating" winter contribution to atmospheric C would be supportive of your statement. It seems that assumptions are being made that current models assume that the wintertime contribution is nil. In fact some models may over estimate this variable. Again, an example of a widely used, modern model that excludes or under represents wintertime production of CO2 would be illustrative.

The referee raises a good point here. Rewording is necessary, as it has proven difficult to come up with a specific example of a widely used, modern model that underrepresents wintertime production. Instead, we can draw our attention to the fact that seasonal variation in soil CO2 fluxes is not always mentioned in meta-analyses of global soil carbon studies (Scharlemann et al., 2014). Though wintertime

measurements may have been incorporated into individual studies, by neglecting this information in a meta-analysis, the reader is left wondering if overwinter CO2 emissions were taken into account at all. Rather than imply that all current models assume that wintertime contribution is nil, we clarify in the manuscript that there is an existing abundance of growing season studies and a general lack of wintertime CO2 soil knowledge, along with continued efforts to include non-growing season/overwinter soil CO2 emission measurements in various models and inventories (Fahnestock et al. 1999, Raich and Potter, 1995).

Section 2.3 Model Development. Line 30. How did you calculate snow pack porosity, and tortuosity? Was snow pack density measured at different intervals or assumed homogeneous for the different "steps"? Also, Fick's 1st Law of Diffusion is adequate for explaining flux in a 1-dimensional, relatively homogeneous medium. However we know that a snowpack stratigraphy is highly variable in space and time. Furthermore, assuming the non-static/non-homogeneous nature of wind and how it affects the snowpack in a very localized manor, could lateral flux occur with the snow pack. Also, elaboration on the role of dense wind slabs, sun crusts, and other ice crusts/lenses within the snowpack would be enlightening. In addition, it seems plausible that Fick's 2nd Law of Diffusion could potentially be useful.

Several assumptions were made for the model and have not been previously stated clearly. As suggested, detail is added to the manuscript to further explain parameters such as snow pack porosity and tortuosity. Snow pack diffusivity values in the model were based off a range of acceptable values to encompass all possibilities in iterations of model runs (at step-change). The step-change snow diffusivity possibilities range from "dense" snow (close to soil diffusivity values) to "light" snow (close to values of CO2 diffusivity in air). With this simplification, we were able to avoid the difficulty of estimating snow pack porosity and tortuosity, as snow pack diffusivity encompasses porosity and tortuosity measurements. Similarly, snowpack density values were not individually calculated or estimated, as snowpack diffusivity also encompasses snowpack density. It is important to note that we did not iterate through a range of snowpack diffusivities for pre-step-change conditions. Simply put, our initial conditions before the simulated advective event varied in snow depth and soil diffusivity, but not snow diffusivity. The referee brings up a critical point with our assumptions in terms of variation in snowpack density in space and time. Yes, our model does assume homogeneous density through vertical space, though "tests" a range of densities by working through a range of step-change snow diffusivities. It is certainly possible that lateral flux could occur within the snowpack, especially with wind slabs, sun crusts, and ice lenses. These physical features likely occurred at our field sites, but are unaccounted for in our modelling—as noted, modelling lateral CO2 transport through a snowpack with this 1-D model is considered impossible. Once we breach the possibility of Fick's 2$^{nd}$ Law of Diffusion, we could be over-complicating the situation for what we were looking to do: understand and observe the differences in diffusive and advective transport through snowpacks, despite the challenges of wintertime measurement. A few studies in the past have used Fick's 2$^{nd}$ Law of Diffusion to model similar events (Solomon and Cerling, 1987), but the overwhelming majority of CO2 diffusive studies use Fick's 1$^{st}$ Law. This is likely because Fick's 2$^{nd}$ Law of Diffusion reduces to Fick's 1$^{st}$ Law of Diffusion when it is simplified and applied to a steady state.

Conclusion. Why is total "accounting" via eddy covariance lacking in this regard? At the outset it would appear that eddy covariance can tell you not only the rate of flux, but the net production of CO2 for a given footprint (accounting?), while eliminating margin for error i.e. snowpack variability. What other sources of CO2 would be accounted for in addition to soil respiration that would not allow you assume all measured net wintertime CO2 was in fact from the soil? A few more sentences explaining your statements/reasoning that in-situ CO2 probes are superior would be enlightening.

Detail added, as suggested. We are not intending to give the impression that total "accounting" via eddy covariance is lacking in this regard. What we are trying to indicate here is that in-situ CO2 probes are not superior to eddy covariance, but are typically cheaper, can be deployed more easily and more frequently, and can give us an indication of what is going on within the snowpack in terms of CO2 transport.

Technical comments

Line numbering appears off, continues from abstract through first portion of introduction, and then switches back mid way. No other technical or grammatical errors were noted.

Thank you.

References

Dyer, J.L. and Mote, T.L.: Spatial variability and trends in observed snow depth over North America. Geophys. Res. Lett. 33(16). 2006.

Fahnestock, J.T., Jones, M.H., and Welker, J.M.: Wintertime $CO_2$ efflux from arctic soils: Implications for annual carbon budgets. Glob. Biogeochem. Cycles, 13(3), 775-779. 1999.

Raich, J.W. and Potter, C.S.: Global patterns of carbon dioxide emissions from soils. Glob. Biogeochem. Cycles, 9(1), 23-26, doi: 10.1029/94GB02723, 1995.

Raich, J.W., Potter, C.S., Bhagawati, D.: Interannual variability in global soil respiration, 1980-94. Glob. Change. Biol., 8, 800-812, 2002.

Scharlemann, J.P.W., Tanner, E.V.J., Hiedere, R., and Kapos, V.: Global soil carbon: understanding and managing the largest terrestrial carbon pool. Carbon Management, 5(1), 81-91, doi: 10.4155/cmt.13.77, 2014.

Solomon, D.K. and Cerling, T.E.: The annual carbon dioxide cycle in a montane soil: Observations, modeling, and implications for weathering. Water Resources Research, 23(12), 2257-2265. 1987.

---

## Author Comment (AC2) · 8 Sep 2017

**Author response to comment on "Explaining CO$_2$ fluctuations observed in snowpacks" by Laura Graham and Dave Risk**

**By Anonymous (Referee 2)**

**(referee comments in black, author responses in blue)**

This work presents both field data and a simple model to address a methodological problem with winter CO2 flux measurements. While both field and model data are presented it isn't really a model-data comparison as the field data isn't directly compared to the model data. As currently structured, there isn't a convincing narrative nor is it clear what is novel. While continuous CO2 datasets are not that abundant, the authors don't present that data and focus on a confusing method of comparing CO2 concentrations to wind speeds. They have adapted a soil diffusion model to the soil-snow system. I'm not sure if the model is too simplistic or if the text just needs greater clarity, but I couldn't follow how this model could help explain the field data. Figure 2 shows that it can create a step change in CO2 concentration gradient, but that isn't enough to be able to match the field data. The paragraph starting on P10 L8 describes how combining modeling and field data could be really powerful, but the temporal changes in the CO2 concentration gradient related to advection that we know happen based on the field data presented here and elsewhere weren't clearly shown. The writing is generally good; my main criticism is that the flow of the narrative, in particular the connections between paragraphs, could be improved. In addition, both the introduction and conclusion sections contain overly broad statements that aren't supported by the rest of the paper. There is certainly need for this type of work and perhaps with some modifications to the model and/or greater clarity of what was done in both the text and the figures could show how the model and field data can be compared, this paper would be acceptable for publication. Finally, I appreciate the authors presenting all the CO2 data. However, it is somewhat surprising to me that snowpack CO2 concentrations could be as low as 151 ppm on Page 6 or well under 400 in Figure 4 or that the atmospheric concentrations was 512 ppm on Page 7, about 100 ppm higher than what it should be. Could the authors justify these seemingly strange measurements?

Thank you kindly. Our goal is that the Biogeosciences community can easily understand this CO2 model-data work, and therefore we appreciate these thoughtful comments by the referee. Through this review process, we hope to clarify the text, improve flow, and solidify how this simple model can be used to help us understand the physical processes of CO2 transport through snowpacks—and not simply generate the CO2 concentrations observed in the field.

Variable concentrations are addressed below with the P6 L34 comment.

P1 L19-21 These lines are way too general for the rest of the paper. The paper is about making the winter flux techniques better not about the soil C and the global C cycle. Or at least it needs to be demonstrated how the results might directly affect the global C cycle.

Introduction altered, as suggested.

P2 L1-3 The authors haven't presented any evidence for yet about why rates might be underestimated

Referee 1 had a similar comment—refer to our reply to Referee 1 for details.

P2 L4-5 This paragraph is just about using the diffusion gradient in the snowpack method to measure soil flux. This method should be explained and its advantages and disadvantages to the other methods should be described.

Detail added, as suggested.

P2 L10-12 This is the key piece of knowledge that this paper is attempting to explain. There is a methodology for measuring CO2 flux that has known limitations. The net result is that it is difficult to separate variability from advection (an artifact) from microbial processes (the actual goal). It would be

very helpful if there was a standard correction that could be used with the diffusion based method to account for advection.

We agree, a standard correction that could be used with the diffusion-based method to account for advection would be very helpful. However, at this point, we must first understand the physical processes of $CO_2$ transport through snowpacks before we can move toward pinpointing a standard correction.

P2 L13-31 Somewhere in this paragraph it needs to be made clear that the assumption is that $CO_2$ production is happening in snow-covered soils, but there are methodological limitations to how the production is quantified. This paper is not about the controls per se, but about how to overcome the methodological limitations so that the mechanisms of $CO_2$ production can be studied.

Thank you for pointing this out. Detail added, as suggested.

P2 L32-34 It would be helpful to talk about of the role of timescale in advective processes in the introduction to justify why you are looking at the hours to days timescale. There is a mention of it in the discussion, but is important here too.

Detail added, as suggested.

P3 L14 Delete the sentence starting with obviously. How do "variable meteorological conditions" affect snow depth?

Sentence deleted, as suggested. Detail added to explain how "variable meteorological conditions" affect snow depth, as suggested.

P3 L16 How long were the measurements made during winter 2014? (as well as winter 2015 in line 34).

Thank you for picking up on this oversight. Detail added, as suggested.

P3 L32-37 I'm confused by this paragraph. In the preceding paragraph, it says that at each of the two stations there were $CO_2$ measurements at 0, 50, and 125 cm, but now it says 25, 50, 75, and 100 cm for NM2. Similarly, in this paragraph, it says the data were collected hourly whereas above it was half hourly. Were the Eosense sensor in addition to Vaisala sensors at the depths above? These sensors need more description. You already said that snow depth and wind speed were measured at both stations in lines 29-31. Somewhere in the methods or results can you indicate what the timing of snow-cover and the maximum depth were.

Detail and clarification added, as suggested. There were some differences and design improvements between the stations for winters 2014 and 2015, which should certainly be made clearer in this paragraph.

P4 L5-7 You need to say why it is important for steady state conditions that snow depth had not changed. Is there a quantitative way that this was determined?

Detail added concerning why it is important that snow depth had not changed. Yes, there was a quantitative way to determine "no change in snow depth". This detail will be added to the updated manuscript.

P4 L9-11 What does "ideal" mean?

"Ideal" in this sense refers to the best set of environmental conditions for which a strong negative correlation between $CO_2$ concentrations and wind speeds could be found.

P4 L18-22 I'm confused about the time period selection. Let's say the snowpack didn't change depth from January 1 to January 5th. Did you look at that whole time period with one regression? In the caption for

figure 2 it says that the average number of 30 minute measurements in the filtered dataset was 29-50 at different heights/sensors. Does that mean the snow depth was changing every day? Or was there some other criteria besides snow depth changing that set the length of time. It seems like the time period selection was based on finding a change in CO2 after an increase in wind speed. Is this assuming that advection is equally affecting the snowpack from the soil surface to the snow surface? Is that a valid assumption? I'm not convinced that this is the right approach to determine the effects of wind on snowpack CO2, but there needs to be a better justification and description of the approach.

The referee's understanding is correct—if the snowpack did not change depth from January 1 to January 5, that entire time period was looked at with one regression. We believe the referee is referring to the caption for Table 2, not Figure 2. The average number of 30-minute measurements is 29-50 at different heights/sensors to effectively indicate "average sample size", and does not necessarily indicate that the snow depth was changing every day. This n indicates the mean number of values (each value is one half-hourly measurements) that was used for each group of regressions (for a given sensor height at each station). Since measurements were recorded half-hourly, we can see how the average duration of each time period ranges from 15 h to 25 h. A potential conclusion from this could be that the snow depth is changing every day—however, there are three distinct criteria that were used for data filtering, as indicated in section 2.2 and in the caption for Table 2. We recognize that this filtering biases our dataset towards having negative relationships between CO2 concentration and wind speed, but this was necessary in order to pick out the advective events we were interested in investigating further.

Is there a different way to do this calculation with fewer assumptions? For example, could you calculate the R2 for the snowpack concentration gradient every 30 minutes and plot that vs. wind speed as a test of whether the wind speed affects the predicted gradient with zero wind? Or compare the concentration gradient to the wind like Seok et al. (2009)? If you examine every time point there is the problem that the previous time points likely affect the relationships. Perhaps you could also try averaging different lengths of time (e.g. 30 minutes to 12 hours or longer)?

The referee presents an interesting suggestion here. However, given the setup of our experiment, it would not be possible to calculate the R2 for the snowpack concentration gradient every 30 minutes. Though it is possible to have higher frequency measurements with the instrumentation we were using, we recorded one value for every 30-minute time interval in order to save battery power—a concern especially when attempting to collect continuous overwinter measurements (with limited solar power, while at a remote location). Evidently, a 1-point regression would not be possible. Though it may be possible to compare concentration gradient to the wind like Seok et al. (2009) with our 2014 data, we reserved this for our 2015 data (Figure 5) when we had more sample heights throughout the snow profile. Though we could try averaging different lengths of time, we are unsure of what purpose that would serve, as we were specifically looking for time periods within our dataset when we would find a negative correlation between wind speed and CO2 concentration within the snowpack.

More detail can be added to this section, especially to indicate that we acknowledge a bias in our data filtering technique.

P5 L6-11 This paragraph needs to be clarified to describe what the model is doing. How are the initial conditions set? What are "step changes in transport rate?" I thought there were step changes in the parameters. Based on figure 2 there is some constant CO2 gradient before time zero and then there is a step change in the diffusivity and the CO2 gradient adjusts quickly. Is there a fixed emission of CO2 from the soil and then the combinations of the diffusion/depth parameters determine the CO2 concentration gradient? Is there a temporary step change and then it returns to the initial value? Or does Storage flux needs to be defined, perhaps even in the introduction. Why is the soil diffusivity included in the model? How is snow depth included? Is the model run with all possible permutations of the 3 parameters?

Detail can be added and this paragraph can be clarified, as suggested. For instance, initial conditions are mentioned in the last two sentences of that paragraph already, but perhaps not clearly stated as initial conditions: "Snow diffusivity before the step change was held constant at 8.06 × 10−6 m2s−1. Each model run began with the system in equilibrium state (with storage flux set to 1 μmol m2s−1)." "Step

changes in transport rate" refers to the step changes in the parameter "snow diffusivity at step change"—this can be easily clarified. The referee is correct: there is a fixed emission of $CO_2$ from the soil in the model, and the changes in parameters determine the $CO_2$ concentration gradient (and therefore calculated storage flux). A definition of storage flux can be added for further clarification. Soil diffusivity was included in the model to determine if $CO_2$ transportation in a diffusive model behaved as expected with an abrupt vertical switch in diffusivity. A description of how snow depth is included is in the paragraph immediately preceding the paragraph in question. Yes, the model was run with all possible permutations of the 3 parameters—this will be clarified both in the text and in the Table 2 caption.

P5 L24-25 What does "the rate at which modelled $CO_2$ responded to an induced wind event" mean? What is the rate? Is an induced wind event the same as an advective wind event in line 15?

The "rate at which modelled $CO_2$ responded to an induced wind event" refers to change in $CO_2$ over time (ppm/s) with the step change in snowpack $CO_2$ diffusivity. Graphically, it is the slope of the line of recovering modelled flux. Yes, an induced wind event is the same as an advective wind event in line 15. Clarification added, as suggested.

P5 L30-31 What is the "enhanced concentration profile experiment?" How were the data processed?

Clarification added, as suggested. The "enhanced concentration profile experiment" is the winter 2015 data. The data "processing" refers simply to how the rate of change of $CO_2$ per unit time after a noticeable wind event was calculated (indicated at the end of the sentence).

P6 L14-19. I'm confused by this example. I thought the ideal situation was when wind speed increased gradually and then abated. These figures show the opposite with wind speeds gradually decreasing and then increasing. Figure 3 seems like a good example, but I'm confused by Figure 4. The snow-atmosphere interface seems hard to measure. Is there a time period with a deeper snowpack that could be shown instead? Why are the $CO_2$ concentrations so much lower (360-390 ppm) than atmospheric (512 ppm)? There was a brief period when the concentration was around 420 ppm that seems to be driving the relationship in this case. If those few hours of data were removed, it doesn't look like there would be much of a relationship.

Clarified, as suggested. For instance, though wind speed increasing gradually and then abating is stated as an ideal situation, the opposite process could be considered similarly ideal—this edit has been made. Figure 3 shows the same time frame, but deeper in the snowpack. Showing these figures side-by-side allows for direct comparison of how the $CO_2$-wind relationship changes with depth into the snowpack. The $CO_2$ concentrations here are considerably lower than the average reported atmospheric concentration, yes. However, there is considerable variability in atmospheric $CO_2$ concentrations throughout the sampling period.

P6 L20-22 Wasn't data collected the whole winter? The data shown in figure 3 and 4 are not included in this time period. This should be clarified in the methods section.

Clarified, as suggested.

P6 L22 How come there is no data for the 0 depth on NM1?

Clarified, as suggested.

P6 L34 How can you get a concentration of 151 ppm $CO_2$ in the snowpack? Is $CO_2$ being consumed or is it some kind of measurement error? Similarly, why is the atmospheric $CO_2$ 512 ppm (P7 L5)? Is this a calibration error.

Detail added, as suggested. Upon further inspection, some values were not filtered out before making the plot. The 151 ppm value is the typical reading when the sensor relay did not turn on. Other than the 151 ppm value, most values are > 290 ppm and sometimes low values can arise if water droplets freeze in the

tubing and a suction creates subambient pressure in the sensor. We have filtered these out as causation is known, and this should have been done before. Thanks to the reviewer for catching this. Average atmospheric CO2 over the sampling period in 2015 is calculated as 512 ppm. There is probably some calibration offset involved here, though sensors were calibrated immediately before the study, and checked afterwards. It is also important to recall that the sensors have an error of several percent. But lastly it is important to remember that those measurements are taken just above the surface and also include stable nighttime measurements when concentrations just above the surface often increase significantly. At some of our sites in Canada, we have observed night concentrations in excess of 700 ppm, as measured with a Picarro CRDS analyzer. For this study, our sites are productive systems with warm winter soils that never freeze because of early snowfall, and so healthy respiration in winter does have an important effect on concentrations just above ground level, and are in this case unbalanced by photosynthetic processes. Oscillations in atmospheric CO2 concentrations with time will have interesting feedbacks to the snowpack concentration profile, which we have seen in modelling, but that's another study.

P7 L5-7 Either test for a relationship between wind speed and CO2 or delete this sentence. It is surprising to me that so much of the first week or so there is no concentration gradient in the snowpack. That is on the 16th-17th and 19th-23rd the whole snowpack is essentially the same as the atmosphere. This seems somewhat surprising. It doesn't seem that much windier than the second week of the experiment. What is happening?

Clarification added, as suggested. For instance, "may have" is changed to "was". Based on the data we collected, it is unclear why the snowpack concentrations from the 16th-17th and 19th-23rd are essentially the same as the atmosphere. Possibilities include ice lenses, temperature changes, or changes in density. Referee 1 had a similar comment—refer to our reply to Referee 1 for details.

P7 L18, L23 What does equilibrium mean? Based on Figure 2 it seems like the model has some step change in parameter and then the CO2 gradient adjusts instantaneously. Similarly, I don't know what a scenario is.

Detail added, as suggested. Equilibrium refers to no change in the modelled storage flux (storage flux at a constant 1 $\mu$mol m2s$^{-1}$). A scenario is a model run under a given set of parameters.

P7 L31-34 I'm not sure why soil diffusivity was included the model as a parameter.

Detail added, as suggested. Explanation also found with P5 L6-11 comment.

P8 L6-13 There are two different phenomena described in this paragraph. One is that there is a monotonic decrease in CO2 from the soil surface to the snow surface if the only source of CO2 production is the soil. I'm sure with ice lenses or if the density of the snowpack isn't constant that there are ways this couldn't be true, but it seems like this should generally be true. The more important question is whether there is a relationship between CO2 concentration and wind speed. The authors have chosen to look over time to see if

Referee 1 also brought up the consideration of ice lenses and variations in density—refer to our response to Referee 1 for further detail. P8 L6-13 comment was left incomplete by Referee 2.

P8 L14-17 You should either calculate storage fluxes or remove this paragraph.

Storage fluxes could perhaps be calculated with this data. However, it is still important to look at concentration gradients before over-complicating our understanding of the physical processes.

P8 L18-26 I'm not sure why you can conclude that advective transport needs to be taken into account by the fact that during 33.6% of the time analyzed there is a relationship between CO2 and wind speed. These two paragraphs don't have much quantitative analysis in them.

The 33.6% value refers to a simple calculation of percentage of wintertime measurements that satisfied all three conditions. This statistic is important especially since the filtering process biased the data we presented. This is clarified, as suggested.

P8 L27-35 Can you give a clearer description of how these processes that occur at different time scales would affect the $CO_2$ concentration gradients and the fluxes measured with Fick's Law? What are "a continuously enhanced friction velocity" and "an enhanced diffusive regime?" It seems like 54 days of measurements should be enough to capture some synoptic variability if you looked for it.

Detail and clarification added, as suggested.

P9 L9-12 I'm not sure I understand exactly what the model did or what equilibrium means. If I look at figure 2 it seems like an instantaneous change in diffusion led to an instantaneous chance in concentration gradient. I don't see any change over time in $CO_2$ concentration which is what I would have thought disequilibrium would be.

Clarification made, as suggested. Figure 2(b) does in fact indicate an instantaneous change in $CO_2$ concentration, along with the instantaneous change in concentration shown in Figure 2(a).

P9 L13-14 I don't understand these sentences.

Sentences clarified in the text. Generally speaking, these sentences are meant to indicate that the diffusive model used can be used to mimic advective events, and that this method is simpler than other models that use a diffusive-advective coupled approach.

P9 L16-27 Why not try to match the model conditions exactly to the field conditions to start at least in terms of $CO_2$ concentration

We tried to start with reasonably close conditions, but did not seek to match exactly because of course we had a broad portfolio of starting concentrations to begin with from the various sites and depletion events. Other studies have applied an iterative procedure to do something similar, and could be applied on a site by site basis if adapted to snow environments—as the referee suggests (e.g. Latimer and Risk, 2016). Though we could have attempted this, our primary goal was to properly understand and illustrate the underlying physics of $CO_2$ transport through snowpacks, and to test whether a simplified enhanced diffusion model could be used—because simplified techniques would make a better basis for iterative field-model fusion applications. As such, matching the model conditions exactly to the field conditions is unnecessary.

P10 L6-7 Just like the beginning of the introduction, this sentence seems like an overreach with no connection to the rest of the text.

Adjustment made, as suggested.

P10 L11-14 This seems like where model data synthesis could really move this field forward.

We agree. However, we believe that we must first have a thorough understanding of the most basic physical processes of $CO_2$ transport within and through the snowpack. There are many other more complicated ways of modelling $CO_2$ transport in various diffusive media—this is a simpler technique to get to the basics of the differences between diffusion and advection of $CO_2$ in snowpacks.

P10 L15-21 Alternative measures of $CO_2$ flux need to be discussed earlier in the manuscript.

Detailed added, as suggested. This is also mentioned in the comment referring to P2 L4-5.

P10 L22-27 This study show snow profile depletions, but I wouldn't say that it explains them. While I agree with the sentiments in the rest of the paragraph, they aren't direct conclusions of the work here.

Text altered, as suggested.

Figure 1. This figure can be deleted. Or it needs to be improved so that the labels match up to the icons and the depths are shown.

Figure improved. Depths are not shown, as the schematic represents two winters with slightly differing sensors depths.

Figure 2. Indicate that an instantaneous change in the diffusivity mimics advection. Storage flux needs to be defined in the text somewhere. Either call it storage flux or apparent storage flux. Indicate the depths on panel b.

Clarifications made, as suggested.

Figure 3. Use the data not the record number on the x axis. Would you expect there to be a hysteresis because of advection? That is, if you drew a line connecting all these points in time would the concentration be higher than average when the winds are decreasing and then lower than average as winds are increasing again? Or maybe vice versa? I realize it is not a crucial question for the model-data comparison, but it seems important for converting concentration gradients into fluxes.

Date used instead of record number on the x-axis, as suggested (applied also to Figure 4). If there is some sort of hysteresis due to advection, it would likely be very hard to distinguish in a time span of hours to days.

Figure 4. Why not pick a time to show when this sensor is really in the snowpack?

A corresponding time when the sensor is "really" in the snowpack is shown in Figure 3. By showing a sensor higher up in the snowpack (closer to the atmosphere) in this figure, we are able to demonstrate a difference in $CO_2$-wind speed relationship with height within the snowpack.

Figure 5. The different colors/dashes are hard to see. It would be better if the wind speeds were in a separate panel. Atmospheric $CO_2$ probably isn't necessary to snow either.

Clarification to the different colours/dashes has been done, as suggested. Atmospheric $CO_2$ is removed, as suggested. We believe wind speeds on the same panel allow for easier direct comparison, even if the figure appears to be confusing at first glance. The wind speeds can be placed in a separate panel, if necessary.

Figure 6. The dashes are hard to distinguish. Why are there 4 cases for a and b but only 2 or 3 for c and d? Are the lines on top of each other? If so, make this clear in the caption.

Dashes altered, as suggested. There are 4 cases for a, b, and d, and only 2 cases for c. Lines are on top of each other in d. The 2 cases for c both refer to short-term storage flux—factor increase in $CO_2$ flux for long-term storage flux is incalculable (0 divided by 0). These clarifications are added to the caption, as suggested.

Figure 7. I like the idea of a conceptual figure, but I'm not sure why lots of little arrows represent diffusion and one big arrow represents advection. Can you make something that shows how the concentration gradient and fluxes change over time in response to wind? Maybe some combination of the information in Figure 2 and Table 3 along with a calculation of the flux using Fick's law and a calculation of the storage flux over time?

A change to this conceptual figure with an "x-axis" demonstrating time will help clarify this confusion with different sized arrows representing diffusion and advection: small, constant movement of $CO_2$ (diffusion)

is represented with small arrows, whereas larger packages of $CO_2$ (advection) moves less frequently and is represented with a larger arrow.

Table 2 is a good summary, but it seems like you can get rid of n as duration is essentially n/2

Though duration is essentially n/2, it is important to show both n and duration. This is because they have different purposes: n gives an indication of the robustness of the $R^2$ measurements, whereas duration gives a more practical visualization of the length of the time periods.

Table 4 Can these events also be shown on figure 5? The measurement depth is in the caption and can be removed from the table.

Thanks for these two suggestions. As space allows, these events are added to figure 5 (there is a lot already going on in Figure 5, but we understand the importance of pointing out the data that was analyzed further). Measurement depth removed from table, as suggested.

---

## Author Response (AR2)

**Author response to comment on "Explaining CO$_2$ fluctuations observed in snowpacks" by Laura Graham and Dave Risk**

**By Anonymous (Referee 3)**

**(referee comments in black, author responses in blue)**

General Comments

Graham and Risk explore carbon dioxide flux through snow in two different snowpacks in Nova Scotia. The focus is on the timescales of CO2 "recovery" (if you will) after wind-induced pressure pumping events (this is similar to the Venturi effect where an increase in speed results in a decrease in pressure).

We thank the referee for their thoughtful comments and suggestions. We have responded to each comment below.

It honestly took me a while to figure out the major points of the manuscript, which was in the timescale of the re-establishment of consistent CO2 gradients in the snowpack after disturbances for the use of gradient-based approaches. This is fine, but wasn't entirely apparent from the introduction, where the findings of Bowling, Massman and others could have been written in a way to point to a more clear overarching question. Doing so will lead in my opinion to a much more compelling manuscript.

To alleviate some of this confusion, we have added some linking statements to the end of the second last paragraph of the introduction (where Bowling and Massman studies are already discussed). The new sentences are: "These studies that investigated advective influence on CO$_2$ transport in snow systems encouraged further study in this area, and so we intended to help fill this gap with our study. To do so, we investigated the mid-range timescale of the re-establishment of consistent CO$_2$ concentration gradients in the snowpack after a wind-induced disturbance using both field and modelling methods."

The text could be more efficient throughout. As a first example, " Datasets have shown that winter..." on line 2 could simply be, "Winter...".

Thank you for this observation. Small edits have been made throughout to streamline the text.

The paper was not particularly well-referenced with a mere 27 references on a topic that has attracted much more attention than this.

We have added a few key references with the advent of this comment. We believe that our paper is methodically cited with references of high quality and relevance.

line 7 of page 2 isn't entirely accurate. 7% of what? Saturated vapor pressure?

We thank the referee for pointing out this oversight. Detail has been added, with the phrase changed to "an increase of approximately 7% in water holding capacity of air per 1˚C warming".

On what basis is the snowpack diffusion approach the most commonly used technique? One may argue that eddy covariance is used much more frequently but also that careful studies of flux through snow using this technique are perhaps a bit lacking.

That particular statement was in reference to the McDowell 2000 paper, referenced later in the paragraph. Since eddy covariance prevalence may have risen over the past two decades, we have edited the statement to: "The

snowpack gradient technique is a commonly used technique […]." A reference to McDowell and Seok has been added, as well.

on page 2 line 34, the major finding of Bowling and Massman is that "enhanced diffusion" is a major mechanism of gas transport through snow, so citing it here isn't the most accurate thing to do.

Thank you for this observation. The citation has been removed.

On page 4, the GMP can be heated as noted, but this impacts the advection when adding heat to a snowpack. It would be forthcoming to estimate the advective flux induced by the GMP sensors for a conservative estimate of their impact on advection. This also creates a risk of melting and freezing snow and encapsulating sensors.

Though the GMP343 sensors do produce some heat, we believe that the amount of heat produced in this context is negligible, given that the sensors were switched on for the minimum amount of time possible, and that our $CO_2$ concentration values were recorded either half-hourly or hourly. Discussed in earlier revision dialogue, our GMP343 sensors were turned off in between measurements to save power, as well as to minimize the potential impacts on snowpack structure (e.g. freezing snow and encapsulating sensors) and gas dynamics (air movement due to heat—convection). The programmed 30 minute cycle for these sensors was: GMP343 on for 10 minutes to "warm up", 1 minute of measurement, and GMP343 off for the remaining 19 minutes of the 30 minute cycle. Additionally, it is important to note that the optics heaters of the GMP343 sensors remained off (default setting) for the entirety of this experiment. Optics heaters should be turned on when there is a risk of condensation on the optics surface. However, in relatively constant temperature environments like snowpack and soils, even if humid, we have found that this (on/off) approach does not render the sensors subject to condensation. Condensation is often more of an issue in atmospheric settings that have quick changes in temperature. As for the power consumption of the sensors, "without optics heating" is < 1 W, and "with optics heating" is < 3.5 W. Having a sensor on for 11 minutes out of every 30 minutes with the optics heating off results in ~9.5 times less heat emitted from sensor power than when compared with having a sensor on continuously for 30 minutes with optics heating on (11min/30min = 0.367, 1 W continuously with optics heating off → 3.5W/0.367W = ~9.5). Yes, the measurement sample was taken during the "warmest" minute of the 30-minute interval—the 11th minute—but we believe that the potential heating impacts were avoided overall. Finally, it is important to note that adding heat to a system of fluids typically induces convection (rather than advection, specifically). Convection is often defined as movement of molecules due to some combination of both diffusion and advection, and so we conclude that estimating the impacts of the minimal GMP343 sensor heating on advective flux is unwarranted.

I'm rather confused by section 2.2 in which it seems like only periods that conform to certain assumptions are chosen, which will bias the results in favor of these conditions and not provide a representative estimate of CO2 flux during the snow-covered period. This topic is covered in the discussion, but led to confusion in the results section.

We assuage this confusion by reiterating the fact that we are not in fact trying to provide an estimate of $CO_2$ flux during the snow-covered period. Clarifications found in the discussion section are now emulated in both section 2.2 and in the results section with statements like: "With the use of our filtering process, our analysis does not represent an estimate of $CO_2$ flux during the snow-covered period," and, "This value does not represent an estimate of the $CO_2$ flux during the snow-covered period, since we used a biased filtering process to identify wind events during periods of steady snow cover."

The 1D model development on page 5 was a bit lacking as the role of temperature, pressure, and tortuosity was ignored or described in a cursory manner. The sensitivity analysis in section 2.4 helps assuage some of these concerns but it should be placed after the model description for continuity.

We thank the referee for the suggestion regarding relocating the "Sensitivity testing" section. It has now been incorporated into section 2.3 (formerly "Model development"), and we hope this clarifies any concerns regarding the more superfluous aspects of the model.

'is considered impossible' is incorrect on page 6, it's just that 3D transport is not modeled in a 1D model.

The referee makes a fair point. The sentence including the mentioned phrase has been adjusted to, "These features are unaccounted for in our modelling, as modelling lateral $CO_2$ transport through a snowpack in addition to vertical transport would require a 3-D model."

'under certain conditions' in the results induces curiosity as to what these conditions are. Do they fall under certain classes or conditions?

Confusion arises here due to the flexible definition of the word "conditions", and if a given reader interprets the word in a mathematical or environmental sense. We've removed this confusion, as we are trying to make the point that wind speed sometimes strongly controlled $CO_2$ concentration within the snowpack (but not always). The reworded sentence states: "Wind speed sometimes had a very strong effect on $CO_2$ concentration within the snowpack (Figs. 3 and 4)."

"These values were a good representation of the CO2 concentration at the snow-air interface." what does this mean?

This statement arises because we are commenting on the usefulness of having $CO_2$ values from where the snow surface is approximately equal to the measurement height (124 cm and 125 cm from the soil surface, respectively). These sentences have been clarified, now stating: "Figure 4 shows measurements at NM1 over the same time period from 125 cm above ground. These $CO_2$ values were very close to predicted atmospheric concentrations, as the average snow depth over this time period at NM1 was 124 cm, very near the measurement height. The closeness of the measurement height to the snow surface indicates these values were likely a good representation of the $CO_2$ concentration at the snow-air interface."

On page 10 line 27 or so, note the findings of Rains et al. (2016, doi: 10.1016/j.coldregions.2015.10.003) that variations in wind speed at low (multiple hour) frequencies may be more effective than variations in atmospheric pressure for explaining changes in snowpack CO2 concentrations. (see also the upper part of page 12 on multi-measurement comparisons).

Thank you for pointing out this comprehensive study by Rains et al. (2016). Detail pertaining to that study has been added to both sections 4.1 (Wind causes short-lived advective anomalies) and 4.3 (Field-model comparisons), as recommended.

in section 4.2, these are essentially the findings of Bowling and Massman (2011) which should be cited.

We thank the referee for this observation. A citation for Bowling and Massman (2011) has been added to section 4.2.

We hope that these minor changes have added to the overall cohesiveness of the test. Thank you for your time, comments, and consideration.

[revised manuscript text omitted]